



# Origin and variability of volatile organic compounds observed at an Eastern Mediterranean background site (Cyprus)

Cécile Debevec.[1,2], Stéphane Sauvage[1], Valérie Gros.[2], Jean Sciare[3,2], Michael Pikridas[3], Iasonas Stavroulas[3], Thérèse Salameh[1], Thierry Leonardis[1], Vincent Gaudion[1], Laurence Depelchin[1], Isabelle Fronval[1], Roland Sarda-Esteve[2], Dominique Baisnée[2], Bernard Bonsang[2], Chrysanthos Savvides[4], Mihalis Vrekoussis[3,5,6], Nadine Locoge[1].

[1] Département SAGE, IMT Lille Douai, Univ. Lille, Lille, 59000, France
[2] Equipe CAE, Laboratoire des Sciences du Climat et de l'Environnement (LSCE), Unité Mixte CEA-CNRS-UVSQ, Gif sur Yvette, 91190, France
[3] Energy, Environment and Water Research Centre, the Cyprus Institute (CyI), Nicosia, 2121, Cyprus
[4] Department of Labour Inspection (DLI), Ministry of Labour, Welfare and Social Insurance, Nicosia, 2121, Cyprus
[5] Institute of Environmental Physics (IUP), University of Bremen, Bremen, 28359, Germany
[6] Center of Marine Environmental Sciences (MARUM), University of Bremen, Bremen, 28359, Germany

*Correspondence to*: Stéphane Sauvage (stephane.sauvage@imt-lille-douai.fr) – Cecile Debevec (cecile.debevec@imt-lille-douai.fr)

**Abstract.** More than 7,000 atmospheric measurements of over sixty $C_2$-$C_{16}$ Volatile Organic Compounds (VOCs) have been conducted at a background site in Cyprus during a 1-month intensive field campaign held in March 2015. This exhaustive dataset consisted in primary anthropogenic and biogenic VOCs, including a wide range of source-specific tracers, and oxygenated VOCs (originating from various origins) that were measured on-line by flame ionization detection/gas chromatography and proton transfer mass spectrometry. On-line submicron aerosol chemical composition was performed in parallel using an aerosol mass spectrometer. This study presents high temporal variability of VOCs and their associated sources. A preliminary analysis of their time series was performed on the basis of independent tracers (NO, CO, black carbon), meteorological data and clustering of air mass trajectories. Biogenic compounds were mainly attributed to be of local origin and showed compound-specific diurnal cycles such as daily maximum for isoprene and nighttime maximum for monoterpenes. Anthropogenic VOCs as well as oxygenated VOCs displayed higher mixing ratios under the influence of continental air masses (i.e. Western Asia) indicating that long-range transport significantly contributed to the VOC levels in the area. Source apportionment was then conducted on a database of 20 VOCs (or grouped VOCs) using a source receptor model. The positive matrix factorization and concentration field analyses were hence conducted to identify and characterize co-variation factors of VOCs that were representative of primary emissions as well as chemical transformation processes. A six-factor PMF solution was selected, namely two primary biogenic factors (relative contribution of 43 % to the total mass of VOCs) for different types of emitting vegetation; three anthropogenic factors (short-lived combustion source, evaporative sources, industrial and evaporative sources, 21 % all together), identified either of local origin but also from more distant emission zones (i.e. the south coasts of Turkey); and a last factor (36 %) associated to a regional background pollution (air



masses transported both from Western and Eastern Mediterranean regions). One of the two biogenic and the regional background factors were found to be the largest contributors to the VOCs concentrations observed at our sampling site. Finally, a combined analysis of VOCs PMF factors with source apportioned organic aerosols (OA) helped to better distinguish between anthropogenic and biogenic influences on the aerosol and gas phase compositions. The highest OA concentrations were observed when the site was influenced by air masses rich in semi-volatile OA (less oxidized aerosols) originating from the Southwest of Asia, in contrast with OA factor contributions associated with the remaining source regions. A reinforcement of secondary OA formation also occurred due to intense oxidation of biogenic precursors.

# 1 Introduction

Ambient air is one of our vital natural resources. However, in addition to having direct or indirect impacts on the environment and climate, air pollution also has adverse health effects, most notably on the respiratory system (Nel, 2005). As a consequence, ambient air pollution has been classified as carcinogenic to humans by the International Agency for Research on Cancer (IARC, 2013).

There are a myriad compounds found in ambient air. Volatile Organic Compounds (VOCs) have been known as one of the principal trace constituents in the atmosphere and they include a large number of species having lifetimes ranging from minutes to months. Their distribution is the result of three combined processes: (1) VOCs are released to the atmosphere by various natural and anthropogenic sources. Emission by vegetation is regarded as the largest natural source on a global scale (Finlayson-Pitts and Pitts, 2000; Guenther et al., 2000). Anthropogenic VOCs are released into the atmosphere by human activities, especially those related to vehicular exhausts, evaporation of gasoline, use of solvents, emissions of natural gas and industrial processes (Friedrich et al., 1999). Although their emissions are quantitatively smaller than biogenic ones at a global scale (Guenther et al., 2000), anthropogenic VOCs can be the most abundant found in urban areas. Once released in the atmosphere, the temporal and spatial variabilities of VOCs are influenced by (2) mixing processes, closely related to meteorological conditions within the mixing boundary layer, which tend to redistribute air pollutants uniformly through advective and convective transport patterns on a regional or long-range scale, especially for long lifetime species (ethane, propane). During their transport, they undergo (3) removal processes or chemical transformations due to atmospheric photo-oxidants such as ozone ($O_3$) during both night and day, nitrate radical ($NO_3$) at night and hydroxyl radical (OH) during daytime (Atkinson, 2000; Atkinson and Arey, 2003). Therefore, in order to perform an exhaustive characterization of VOCs sources, it is important to know their chemical composition but also their potential intensity variations on different time scales.

A robust tool to identify emission sources is Positive Matrix Factorization (PMF - Paatero, 1997; Paatero and Tapper, 1994) is one of the various tools developed to identify emission sources. Over the last decade, this source-receptor approach has been intensively used in urban areas (Baudic et al., 2016; Brown et al., 2007; Gaimoz et al., 2011; Latella et al., 2005; Salameh et al., 2016; Yuan et al., 2012) but also in remote/rural environments (Lanz et al., 2008; Leuchner et al.,





2015; Michoud et al., 2017; Sauvage et al., 2009). Other receptor models such as Principal Component Analysis/Absolute Principal Component Scores (PCA/APCS) (Chan and Mozurkewich, 2007; Guo et al., 2007), UNMIX (Jorquera and Rappenglück, 2004; Olson et al., 2007) and Chemical Mass Balance (CMB) (Badol et al., 2008; Na and Pyo Kim, 2007) have been previously used in VOCs source apportionment. Although both models have aims similar to that of PMF, they have different mechanisms and each approach has advantages and limitations as described in several comparison studies (e.g.Anderson et al., 2002; Miller et al., 2002; Song et al., 2008; Viana et al., 2008; Willis, 2000). Studies show that PMF models are more efficient in identifying source profiles (Jorquera and Rappenglück, 2004; Miller et al., 2002). Yuan et al., 2012 stressed the importance of different reactivity of VOCs and the impact of photochemical aging on the interpretability of the resolved factors as source profiles that have not been considered in most of the studies applying PMF. Therefore, in remote/rural environments, despite the assumption of non-reactivity or mass conservation (Hopke, 2003), PMF can be either used to identify aged primary sources or to get insights about the sources and processes involved in the evolution of measured VOCs (Michoud et al., 2017; Sauvage et al., 2009).

VOCs are also key players in photochemical processes leading to secondary pollutant formation, such as tropospheric $O_3$ (Seinfeld and Pandis, 1998) and secondary Oxygenated (O)VOCs (Atkinson, 2000; Goldstein and Galbally, 2007; Seco et al., 2007). Undergoing multigenerational oxidation processes, these secondary OVOCs react with atmospheric oxidants leading to increasing functionalized products with sufficiently low volatility (Aumont et al., 2012; Jimenez et al., 2009; Kroll and Seinfeld, 2008) to take part in the formation of Secondary Organic Aerosols (SOA) by nucleation or condensation onto pre-existing particles (Fuzzi et al., 2006; Kanakidou et al., 2005). Furthermore, numerical simulation models that take into account chemical transformations suggest that the secondary OVOCs are still reactive and multi-functionalized several days after emission, allowing transport over long distances and thus affecting the oxidant budget as well as the formation of ozone and SOA at remote locations (Aumont et al., 2005; Madronich, 2006). It is, therefore, essential to understand the sources and fate of VOCs in the atmosphere, and especially its evolution during long-range transport.

Affected by important pollution sources, the Mediterranean is a sensitive region affected by both particulate and gaseous pollutants. This air pollution is the combination of (1) long-range transported polluted air masses originating from 3 continents (Europe, Asia and Africa – e.g. Lelieveld et al., 2002; Pace et al., 2006; Ziv et al., 2004) with (2) local emissions either of anthropogenic origin, associated to local human pressure from the surrounding industrial and densely populated coastal areas of the basin (Im and Kanakidou, 2012; Kanakidou et al., 2011) or of natural emissions (Kalogridis et al., 2014a; Liakakou et al., 2007; Owen et al., 2001) and forest fires (Alves et al., 2010; Bougiatioti et al., 2014). Additional aggravating factors are among others: (3) climatic conditions favoring the development of photochemical processes (Fountoukis et al., 2011), (4) the scarcity of precipitation scavenging and (5) a tendency to accumulate pollution due to poor ventilation rates. As a consequence, particulate or ozone concentrations are generally higher in the Mediterranean region than in most continental European regions especially during summertime (Doche et al., 2014; Menut et al., 2015; Nabat et al., 2013; Safieddine et al., 2014). The Mediterranean region is also considered as a prominent climate change "hot spot", considering



interactions between climate and air quality, and is expected to undergo marked warming and drying in the 21$^{st}$ century (Giorgi, 2006; Kopf, 2010; Lelieveld et al., 2014). However, air pollution in this region remains difficult to assess and characterize mostly because of a lack of atmospheric measurements. Additional information on the chemical composition of the air, including the speciation and the reactivity of VOCs, will further improve our current understanding of the complexity

of the Mediterranean atmosphere.

In this context, the ChArMEx (the Chemistry-Aerosol Mediterranean Experiment, http://charmex.lsce.ipsl.fr) (Dulac, 2014) international project of the multidisciplinary regional research program MISTRALS (Mediterranean Integrated Studies at Regional and Local Scales; http://mistrals-home.org) aims at developing and coordinating regional research actions for a scientific assessment of the present and future state of the atmospheric environment in the

Mediterranean basin, and of its impacts on the regional climate, air quality, and marine biogeochemistry.

The aim of this work is to provide a better characterization of the sources and fate of VOCs impacting the Eastern Mediterranean region, focusing on a comprehensive high time resolution detailed chemical composition measured at a representative receptor site. To reach this objective, on-line measurements of a large number of VOCs, including alkanes, alkenes, alkyne, aromatics compounds and OVOCs, have been performed at the Cyprus Atmospheric Observatory (CAO,

http://www.cyi.ac.cy/index.php/cao.html), a receptor site located in Cyprus, in March 2015 within the framework of ChArMEx and ENVI-Med "CyAr" (Cyprus Aerosols and gas precursors) programs. The period under investigation offered contrasted conditions in terms of air mass transport (geographical origin, fast/low transport) and weather (temperature, humidity, precipitations...). CAO offers ideal experimental conditions to monitor long-range transported clean/polluted air masses from the 3 surrounding continents. It is ideally located, close from the coast and far from major Cypriot

anthropogenic areas. The site is also surrounded by widespread vegetation with intense local biogenic emissions making possible the investigation of biogenic and anthropogenic interactions on the air mass composition. In the framework of the ChArMEx program, the Western Mediterranean region was already characterized during an intensive field campaign conducted in a French rural site (Cape Corsica) in summer 2013 (Kalogridis, 2014; Michoud et al., 2017). Our study performed in the Eastern Mediterranean will therefore offer a unique opportunity to provide a comprehensive

characterization of VOCs for the entire Mediterranean.

First, the sampling site is described in Sect. 2 along with the analytical techniques and the computational tools for identifying and characterizing the major sources of VOCs. In Sect. 3, we investigate VOCs levels and their temporal variations in comparison with air quality indicators, meteorological parameters and air masses origins. An accurate identification of PMF factors is proposed in the Sect. 3.5. Then, in Sect. 4.1, we compare VOCs concentrations measured

during this intensive field campaign with previous measurements performed at an another background site of Cyprus but also factor contributions to source apportionment of VOCs measured at a French remote site in Corsica. Finally, the PMF VOCs factors are used in Sect. 4.2 to assess the origin (anthropogenic/biogenic) of non-refractive organic aerosols in PM$_1$ (Particulate Matter with aerodynamic diameter below 1 µm) measured in parallel.



## 2 Material and Methods

### 2.1 Sampling site

Cyprus is an island located on the Eastern part of the Mediterranean Sea, 110 km southerly from the Turkish coast, c.a. 250 km westerly from Lebanon and Syria and 780 km easterly from Crete (Greece). This island covers an area of 9,250 km$^2$ and
includes 648 km of coastline. The main agglomerations of the island are namely, the capital Nicosia, Limassol, Larnaca, Paphos, Famagusta and Kyrenia (321,816; 176,600; 84,591; 61,986; 50,265 and 33,207 inhabitants, respectively, census 2011 – Fig. 1). Air masses circulating over Cyprus are constrained by two mountain complexes, the Troodos and the Kyrenia Mountains (located in the center and the north of Cyprus, respectively).

Within the framework of two French research programs, the Chemistry and Aerosol Mediterranean Experiment
(ChArMEx) and ENVI-Med "CyAr" (Cyprus Aerosols and gas precursors), an intensive field campaign has been performed at a background site of Cyprus (CAO, 33.05° E - 35.03° N, 532 m above sea level, a.s.l. - Sciare, 2016) from 1 March to 29 March 2015. CAO is operating under ACTRIS, the European Research Infrastructure fir the observation of Aerosol, Clouds and Trace gases (http://actris2.nilu.no/). The station is co-operated by the Department of Labour Inspection (DLI) within the network of the "Co-operative programme for monitoring and evaluation of the long-range transmission of air pollutants in
Europe" (EMEP). Therefore, criteria defined by the EMEP and ACTRIS networks provide a high quality assurance for the atmospheric measurements performed at CAO. The station is located in the central area of the island about 20 km from the coast and more than 35 km of main Cypriot agglomerations, with very poor influences of these anthropogenic emission areas. CAO is situated beside a house of the Cyprus Department of Forests potentially entailing a local pollution (typical vehicle circulation during week days). Besides, the measurement site is surrounded by widespread vegetation such as
"maquis", shrubland typical of Mediterranean areas, and oak and pine forests (Fall, 2012), including huge emitters of biogenic (B)VOCs (Owen et al., 2001).

### 2.2 Experimental Set-up

Section SI-1 in the supplement details the temporal coverage of each instrument deployed during the intensive field campaign and presented in this section.

### 2.2.1 VOCs measurements

Non-Methane HydroCarbons (NMHCs) and Oxygenated (O)VOCs, were measured using complementary on-line techniques. The inlets were approximately 3 m above ground level (a.g.l.). Table 1 summarizes the characteristics of the methods performed during the campaign and provides a list of the monitored VOCs.

**GC-FID:**

Two portable automated gas chromatographs (GCs, Chromatotec, Saint-Antoine, France) equipped with a flame ionization detector (FID) were used for on-line ambient air measurements of C$_2$ to C$_{10}$ anthropogenic VOCs and C$_{10}$ biogenic



(B)VOCs. The first analyzer, ChromaTrap, allowed the measurement of $C_2$-$C_6$ hydrocarbons (alkanes, alkenes and alkynes) and the second, AirmoVOC $C_6$-$C_{12}$, the measurement of $C_6$-$C_{10}$ hydrocarbons (alkanes, monoterpenes and aromatic compounds). For each analysis of 30 min, 180 mL for ChromaTrap and 1,320 mL for AirmoVOC $C_6$-$C_{12}$ of ambient air was drawn by an external pump through a 0.315 mm diameter 5-m length stainless-steel line into the system with a flow rate of

18 mL.min$^{-1}$ for ChromaTrap, and 60 mL.min$^{-1}$ for AirmoVOC $C_6$-$C_{12}$. A stainless-steel filter (pore diameter 0.5 μm, Swagelok) was settled at each inlet to avoid aerosols and other fragments entering the system. Each measurement started with an analysis period, 15 min for ChromaTrap and 19 min for AirmoVOC $C_6$-$C_{12}$, and ended with a sampling period, 10 min for ChromaTrap and 22.5 min for AirmoVOC $C_6$-$C_{12}$. Therefore, instruments were synchronized in order to sample air simultaneously.

Regarding AirmoVOC $C_6$-$C_{12}$ analyzer, VOCs were collected by a glass trap filled with adsorbents (Carbotrap B) and maintained at ambient temperature. After sampling, the trap was heated rapidly to 380°C for 2 min so that VOCs were thermally desorbed and directly injected into a separating column (MXT30CE, 30 m × 0.28 mm diameter) located inside the heated oven of the GC. The oven temperature rose from 38°C to 50°C with a constant rate of 2 °C min$^{-1}$, then with a constant rate of 10 °C min$^{-1}$ to 80 °C and finally with a constant rate of 15 °C min$^{-1}$ until the temperature reached 200 °C at the end of

the analysis. An in-depth description of the ChromaTrap analyzer, sampling set up and technical information (pre-concentration, desorption-heating times, type of traps and column types) can be found in Gros et al., 2011. In this study, the oven temperature rose from 35 °C to 202 °C with a constant rate of 15 °C min$^{-1}$.

During the campaign, a gas standard containing a mixture of 30 $C_2$-$C_{10}$ VOCs at a concentration level of 4 ppb (parts per billion by volume) certified by NPL (National Physical Laboratory, Teddington, Middlesex, UK) was injected

three times. These series of measurements allowed checking compound retention times, testing the reproducibility of each instrument and calculating mean response factor per VOC and per analyzer which was used to calibrate the measurements. The calibration measurements have shown that the analyzer reproducibility (expressed here as a mean coefficient of variation of the areas obtained) remained stable during the measurement period under investigation with variations within ± 5 % for ChromaTrap (except for ethylene and i-butane: ± 9 % and ± 7 %, respectively) and within ± 2 % for AirmoVOC $C_6$-

$C_{12}$. At least, two zero gas measurements were performed consecutively before each calibration to check compound blank values. Detection limits have been determined as 3 σ of the baseline and very satisfactory detection limits were found (below 0.2 μg.m$^{-3}$ and 0.1 μg.m$^{-3}$ for ChromaTrap and for AirmoVOC $C_6$-$C_{12}$, respectively).

Uncertainties related to the concentration of each compound have been estimated. As a first step, the main sources of uncertainties were identified partly based on a quantitative expression relating the value of the measurand to the

parameters on which it depends. Each uncertainty arising from these sources was quantified separately and standard uncertainties were combined following the error propagation law. Finally, the expanded uncertainty includes a proportional term to the measurand and a threshold uncertainty (EURACHEM - Ellison et al., 2000) as:

$$U(X) = k \times u(X) + \frac{DL_X}{3},$$      (1)





$U(X)$ is the expanded uncertainty of X, $k$ is the chosen coverage factor (2 here), $u(X)$ the combined uncertainty of X and $DL_X$ is the detection limit of the species X.

In practice, compounds were quantified following the expression given in ACTRIS Measurement Guidelines VOC, 2014. Identified sources of uncertainties included systematic integration errors (in sample, calibration gas and zero gas

measurements), variability of blank values of VOCs determined in zero gas measurements, reproducibility and linearity errors of the sampling system and uncertainty in the standard gas mole fraction. Relative uncertainties of VOCs measured with ChromaTrap analyzer varied from 14 % (ethane) and 73 % (propene) and from 18 % (benzene) and 53 % (o-xylene) for VOCs measured with the AirmoVOC $C_6$-$C_{12}$. Note that compounds measured with the Chromatrap system have shown slightly higher relative uncertainties than ones measured with the AirmoVOC $C_6$-$C_{12}$ due to their respective blank value

which had strong contributions of their variability and integration errors in sample and zero gas measurements (concerned mostly propene, i-pentane and n-pentane).

**PTR-QMS:**

An on-line high-sensitivity Proton Transfer Reaction - Quadrupole Mass Spectrometer (PTR-QMS, Ionicon Analytik GmbH, Innsbruck, Austria) was used for real-time VOCs measurements (e.g. methanol, acetone, isoprene and

monoterpenes). Because this instrument has widely been described in recent reviews (Blake et al., 2009 and references therein), only a brief description of analytical conditions related to ambient air observations is given here.

Ambient air was sampled through a 5-m long Teflon line (inner diameter: 0.125 cm) with a flow rate of 80 mL.min$^-$$^1$ to minimize residence time to 4 s inside the sampling line. A filter (pore diameter: 0.5 μm, Swagelok) was installed at the head of the inlet. The PTR-MS was operating at standard conditions: a drift pressure, temperature and voltage of 2.2 mbar,

60 °C and 600 V, respectively, leading to an E/N ratio (corresponding to the electrical field E, in V.cm$^{-1}$, on the buffer gas number density N, in molecule.cm$^{-3}$) around 130 Townsend. VOCs measurements were performed in a full-scan mode and enabled to browse a large range of masses (m/z 30.0 amu – m/z 137.0 amu). With a dwell time of 5 s per mass, signals of every masses were acquired every 10 min and were then normalized to 1 million primary ions as proposed by Warneke et al., 2001. Eleven protonated target masses were considered here (detailed in Table 1) and other masses were used for the

assessment of the PTR-MS performances (Dolgorouky, 2012) or as independent parameters.

The PTR-MS background for each mass was daily monitored by sampling zero air generated with ambient air passed through a catalytic scrubber that efficiently removed VOCs via a Gas Calibration Unit (GCU, Ionicon Analytik GmbH, Innsbruck, Austria). Daily background values were then averaged, normalized, linearly interpolated onto all other measurements and subtracted from the atmospheric data before final conversion to mixing ratios.

Three calibrations were performed (before, during and after the intensive field campaign) with the GCU and a standard gas mixture provided by Ionicon and containing seventeen VOCs at 1 ppm (parts per million by volume). Instrument response for species targeted was determined through the measurement of a standard gas previously diluted to mixing ratios in the range 1 ppb – 16 ppb. Gas calibrations allowed to determine the reproducibility of measurements, expressed here as a mean coefficient of variation. This coefficient was less than 5 % for all targeted compounds underlining



a good stability of the instrument. Then, targeted masses were then quantified following the expression given in Taipale et al., 2008. For chemicals not found in the calibration gas, their respective ion count rate was directly converted into concentration unit using transmission curve coefficients (de Gouw and Warneke, 2007; Taipale et al., 2008). The latter was determined using the transmission tool of the PTR-MS Viewer software (Version 3.1.0.28, Ionicon Analytik GmbH) and

5 calibration results. Transmission coefficients of masses not calibrated were obtained interpolating known transmission coefficients according to a cubic Hermite spline (Taipale et al., 2008). Detection limits of targeted compounds have been determined as 3 σ of the blank variation and were all below 0.1 µg.m$^{-3}$.

Identified sources of uncertainties were namely, the variability of the ion count rates in zero gas and calibration measurements combined with the dilution, linearity and reproducibility errors of the sampling system, the uncertainty in the

10 standard gas mole fraction and the relative humidity effect on the instrument sensitivity. Relative uncertainty of VOCs measured with PTR-MS varied from 18 % (acetone) to 44 % (methanol). Uncertainties of VOCs concentrations of the gas standard significantly contributed for each targeted species. Additional important contributions were linearity error and detection limit especially for methanol, acetonitrile, isoprene, Methyl Vinyl Ketone (MVK) + methacrolein (MACR) and C$_8$-aromatics.

**Off-line VOCs measurements:**

More than 400 off-line 3h-integrated air samples were collected by active sampling on sorbent cartridges (multi-sorbent and DNPH cartridges). Thirty-nine C$_5$-C$_{16}$ anthropogenic VOCs (including alkanes, alkenes, and aromatics) and 9 BVOCs, as well as 6 C$_6$-C$_{11}$ n-aldehydes were collected by multi-sorbent cartridges, analyzed after by GC-FID, and 10 C$_1$-C$_8$ carbonyl compounds (e.g. acetone and acetaldehyde) were sampled in parallel on the DNPH cartridges. Samples were

20 after analyzed by GC-FID (for multi-sorbent cartridges) and by HPLC-UV (High-Performance Liquid Chromatography coupling with Ultra Violet detection – for DNPH cartridges). These technics have already been described by Detournay et al., 2011 and its set up on the field discussed by Detournay et al., 2013 and Ait-Helal et al., 2014. Here, VOCs measured off-line were used as independent parameters and they will be further presented and investigated in another paper dedicated on biogenic and oxygenated VOCs.

**VOCs intercomparison:**

Benzene and toluene were two compounds measured by different on-line techniques. Therefore, they were chosen to cross-check the quality of the results during the campaign. Benzene and toluene measured by PTR-MS and GC-FID showed good correlation coefficients (r = 0.75 and 0.79 for benzene and toluene, respectively, n = 1,172) with slopes close to one for both compounds (0.93 for benzene and 0.87 for toluene) and low intercepts (0.10 µg. m$^{-3}$ for benzene and 0.13

30 µg. m$^{-3}$ for toluene). The results from benzene and toluene presented in this paper originated from GC-FID measurements since these instruments were used during longer time than PTR-MS (January 2015 – February 2016). Although the GC-FID method employed can also monitor isoprene, yet results presented here originated from PTR-MS due to better sensitivity and analytical performance.



The sum of pinenes measured by the GC-FID was compared to the monoterpenes (non speciated) measured by PTR-MS, yielding the same variability and consistent ranges of concentrations. The same conclusion was obtained for acetone and acetaldehyde, measured by PTR-MS and off-line technique.

As a result, recovery of the different techniques, regular quality checks and uncertainty determination approach have allowed to provide a good robustness of the dataset.

### 2.2.2 Ancillary gas measurements

A large set of real-time atmospheric measurements was carried out by the DLI at the CAO, in order to characterize trace gases ($NO$, $NO_2$, $O_3$, $CO$ and $SO_2$). They are described in more details by Kleanthous et al., 2014. The time resolution was 5 min for each analyser but the presented results were provided only on hourly average.

### 2.2.3 Aerosol measurements

The chemical composition of non-refractory submicron aerosol ($NR-PM_1$) has been continuously monitored using a Quadrupole Aerosol Chemical Speciation Monitor (Q-ACSM, Aerodyne Research Inc., Billerica, Massachusetts, USA), which has been described in detail by Ng et al., 2011. This recent instrument shares the same general structure with the Aerosol Mass Spectrometer (AMS) but has been specifically developed for long-term monitoring. The Q-ACSM instrument was operating non-stop with 30-min time resolution during the whole duration of the campaign totalizing 1,292 valid data points (corresponding to a time recovery of 95 %). The ACSM dataset was validated against $PM_1$ chemical composition results provided by integrated daily resolution measurements. Instrument settings, field operation, calibration and data processing are those reported in Petit et al., 2015.

Black carbon (BC) was calculated using the 880 nm channel of a 7-wavelength (370, 470, 520, 590, 660, 880 and 950 nm) Aethalometer (AE31 model, Magee Scientific Corporation, Berkeley, CA, USA) with a time resolution of 5 min. Assuming difference in the absorption angstrom exponent between fossil fuel and biomass burning derived aerosol, the BC origin was distinguished (Sandradewi et al., 2008).

Real time measurements of particle mass concentrations ($PM_{2.5}$ and $PM_{10}$) were obtained using a Tapered Element Oscillating Microbalance (TEOM 1405-DF, Thermo Scientific, Waltham, Massachusetts, USA) (Clements et al., 2016; Patashnick and Rupprecht, 1991). The time resolution of TEOM analyser was 15 min but the presented results were provided only on hourly average.

### 2.2.4 Meteorological measurement and air-mass origins classification

Meteorological parameters (temperature, pressure, relative humidity, wind speed, wind direction and radiation) were measured every 5 min with a Campbell Scientific Europe (Antony, France) weather station during the whole campaign.

Classification of air-mass origins has been based on the analysis of the retroplumes computed by the Flexpart lagrangian model (Stohl et al., 2005) using CAO as the receptor site. The Flexpart model calculates trajectories of user-





defined ensembles of particles released from three-dimensional boxes. The classification was based on hourly resolution model simulations going back in time 5 days, taking into account only the lowest 100 m a.g.l. (footprint plots), even though the 3 km was modeled. The classification of these backward retroplumes involved 88 source regions, similar to Kleanthous et al., 2014, identified by a custom-made algorithm combined with visual inspection. The source region map is provided in

Fig. 2 based on the residence time of particles over each source region.

**2.3 Identification and contribution of major sources of VOCs**

**2.3.1 Source apportionment by Positive Matrix Factorization (PMF)**

PMF is a multivariate factor analysis tool for identifying and characterizing the "p" independent sources of "n" compounds sampled "m" times at a receptor site. A more detailed presentation of the PMF mathematical theory is described elsewhere

(Paatero and Tapper, 1994; Paatero, 1997). Briefly, the method consists in decomposing a chemically speciated sample data into species profiles of each source identified (representing the repartition of each species to each factor) and the amount of mass contributed by each factor to each individual sample (representing the evolution in time of each factor). The principle can be summarized as follows:

$$x_{ij} = \sum_{k=1}^{p} g_{jk} \times f_{ki} + e_{ij} = c_{ij} + e_{ij} ,$$  (2)

where $x_{ij}$ is the i$^{th}$ species measured concentration (in µg m$^{-3}$ here) in the j$^{th}$ sample, $f_{ki}$ the i$^{th}$ mass fraction from k$^{th}$ source, $g_{jk}$ the k$^{th}$ source contribution of the j$^{th}$ sample, $e_{ij}$ the residual resulting of the decomposition and $c_{ij}$ the reconstructed concentration of the species. The solution of Eq. (2) is obtained iteratively by minimizing the residual sum of squares Q given by Eq. (3):

$$Q = \sum_{i=1}^{n} \sum_{j=1}^{m} \left( \frac{e_{ij}}{s_{ij}} \right)^2 ,$$  (3)

with $s_{ij}$, the extended uncertainty (in µg m$^{-3}$) related to the i$^{th}$ species concentration measured in the j$^{th}$ sample. PMF tool requires a user-provided uncertainty matrix built upon the procedure described by Polissar et al., 1998 to weight individual points. Results are also obtained using the constraint that no sample can have significantly negative source contributions.

**2.3.2 VOCs dataset**

In this present study, EPA PMF 5.0 using a multilinear engine ME-2 (Paatero, 1999) and additional guidance on the use of

PMF (Norris et al., 2014) were used to analyze the VOCs concentrations observed at CAO. Only on-line measurements from 1 March to 29 March 2015 are taken into account in this factorial analysis. The dataset contains a selection of 20 hydrocarbons species and masses divided into nine compounds families: alkanes (ethane, propane, i-butane, n-butane, i-pentane, n-pentane, 2-methylpentane), alkene (ethylene), alkyne (acetylene), diene (m/z 69), terpenes (pinenes - sum of α-pinene and β-pinene), aromatics (benzene, toluene, m/z 107), alcohol (m/z 33), carbonyl compounds (m/z 45, m/z 59, m/z



71, m73) and nitrile (m/z 42). We chose m/z 107 results, measured by PTR-MS, instead of $C_8$-aromatics results, measured by GC-FID, due to their respective uncertainties. Note that, same PMF solutions were found if we considered m/z 137 instead of the sum of pinenes. The final chemical database consisted of 1,179 atmospheric data points with 30min-time resolution and covering 87% of the sampling period. PTR-MS observations have been averaged to correspond to GC-FID time resolution. Input information is summarized in Table 2. Additionally, the PMF approach, regarding data processing and analysis of the quality of the VOCs dataset, is based on the recommendations of Norris et al., 2014 and is presented in the supplement material (Sect. SI-2).

### 2.3.3 Optimized PMF solution

Several base runs with different numbers of factors were performed initially and examined in order to determine the optimal number of factors following the protocol proposed by Sauvage et al., 2009. The selection of the optimal solution relies on several statistical indicators and the physical meaning of factor profiles. The non-negativity constraint alone is not sufficient to produce a unique solution generally. Rotating a given solution and evaluating the rotated results with the initial solution is one approach to reduce the number of solutions. Thus, the second step consisted in exploring the rotational freedom of the PMF solution kept by varying the $F_{peak}$ parameter (Paatero et al., 2002; Paatero et al., 2005) in order to determine an optimized final solution. Therefore, a six-factor PMF solution has been selected and a $F_{peak}$ parameter of 0.8 allowed a better decomposition of the chemical dataset with a low change of the Q-value, consistent with recommendations of Norris et al., 2014.

Quality indicators from the PMF application have been summarized in Table 2. The ratio Q(robust)/Q(true) reached a value around 1.0 indicating that the modeled results were not biased by peak events. The PMF model results explained on average 98 % of the total concentration of the 20 VOCs. Individually, almost all the chemical species also displayed both good determination coefficients and slopes (close to 1) between predicted and observed concentrations, with the exception of species declared as weak but also propane, acetylene, toluene and acetonitrile. Therefore, the limitations of the PMF model to simulate these species should be kept in mind when interpreting the PMF results.

Rotational ambiguity and random errors in a PMF solution were evaluated using DISP (displacement) and BS (bootstrap) error estimation methods (Norris et al., 2014; Paatero et al., 2014; Brown et al., 2015). No factor swap occurred in the DISP analysis results, thus indicating that the 6-factor PMF solution was sufficiently robust to be used. Bootstrapping was then carried out, executing 100 runs, using a random seed, a block size of 97 samples and a minimum Pearson correlation coefficient of 0.6. All the modeled factors were well mapped over at least 85 % of runs, hence highlighting their reproducibility.



## 2.4 Geographical origins of VOCs

### 2.4.1 Local pattern of VOCs sources using Conditional probability function (CPF)

The geographical locations associated with VOCs sources can be approached using CPF (Ashbaugh et al., 1985) as performed in several studies (Gaimoz et al., 2011; Kim et al., 2003, 2005; Xiang et al., 2012; Xie and Berkowitz, 2006, 2007). Source contributions were combined for that purpose with wind direction values measured on site in order to identify the geographical orientation of sources. Specifically, the CPF is defined as:

$$CPF = \frac{m_{\Delta\theta}}{n_{\Delta\theta}} , \qquad (4)$$

where $m_{\Delta\theta}$ is the number of samples in the wind sector $\Delta\theta$ that exceeded the threshold criterion and $n_{\Delta\theta}$ is the total number of samples in the wind sector considered. As done by Kim et al., 2003, relative contributions from each source to the total of all sources was used to minimize the effect of atmospheric dilution. In this study, wind directions were binned into 8 sectors of 45° and the threshold was set at the upper 75[th] percentile of all relative contributions for each source to measure the frequency that a peak occurred under a specific wind direction. Calm winds with speeds of less than 1 m.s[-1] were excluded from the calculations, which represented 8 % of the total observations. Additionally, the CPF analysis makes sense only if air arriving at a receptor site has traveled with a relatively straight path from the source. While this assumption is probably true for receptors and sources that are quite close to each other, it is not generally the case in particular for background receptor sites which are far from local sources. Therefore, CPF has shown the likely local orientation of potential sources and straight paths can be selected using 6h-backtrajectories to simplify the analysis (Zhou et al., 2004; Xie and Berkowitz, 2006).

### 2.4.2 Identification of potential regional emissions areas using Concentration Field (CF)

Since wind measured at a receptor site is not necessarily representative of the initial origin of the air mass, source contributions were then coupled with back-trajectories to investigate potential long-distance pollutions advected to the sampling site. In this study, we have used the CF approach developed by Seibert et al., 1994. This method consists in redistributing concentrations of a variable observed at a receptor site along the back-trajectories, ending at this site, inside a gridded map. Attributed concentration in a cell is then weighted by the residence time that air parcels spent in the cell following Eq. (5) (Michoud et al., 2017):

$$\log(\overline{C_{IJ}}) = \frac{\sum_{L=1}^{M} \log(C_L) \times n_{ij-L}}{\sum_{L=1}^{M} n_{ij-L}} = \frac{1}{n_{ij}} \sum_{L=1}^{M} \log(C_L) \times n_{ij-L}, \qquad (5)$$

with $\overline{C_{ij}}$ the attributed concentration of the ij[th] grid cell, $C_L$ the concentration measured at the receptor site when the back-trajectory $L$ reached it, $n_{ij-L}$ the number of points of the back-trajectory $L$ which fall in the ij[th] grid cell and $n_{ij}$ the number of points of the total number of back-trajectories $M$ associated to the cell concerned.





The 5-day back-trajectories used, along with meteorological parameters (i.e., precipitation), were calculated every 1 hour using a PC-based version of HYSPLIT lagrangian model (version 4.4 revised in February 2016; Draxler and Hess, 1997; Stein et al., 2015). The altitude of the end point was set at 800 m a.s.l., respectively. The constant time step between each point of a back-trajectory is 1 hour.

CF applied to PMF results were performed with the ZeFir tool (version 3.02; Petit et al., 2017). As done by Bressi et al., 2014, wet deposition was estimated by assuming that precipitation (0.1 mm) will clean up the air parcel. The $log(C_L)$ value was set to 0 for all air parcels of the back-trajectory path before precipitation occurred. Back-trajectories have been also interrupted when the altitudes of air masses exceeded 1,500 m a.s.l. to get rid of the important dilution affecting the air masses when reaching the free troposphere (Michoud et al., 2017). Furthermore, in order to consider air parcel with a good

representativity, a weighing function has been implemented to remove high CF uncertainties usually linked with low $n_{ij}$ values. The method implemented in Zefir was based on a "trajectory density" $log(n_{ij} + 1)$ (Bressi et al., 2014; Waked et al., 2014). Indeed, weighing function is empirically determined as:

$$W = \begin{cases} 1.00, \; for \; n_{ij} \geq 0.85 \, max\big(log(n_{ij} + 1)\big) \\ 0.725, \; for \; 0.85 \, max\big(log(n_{ij} + 1)\big) > n_{ij} \geq 0.6 \, max\big(log(n_{ij} + 1)\big) \\ 0.475, \; for \; 0.6 \, max\big(log(n_{ij} + 1)\big) > n_{ij} \geq 0.35 \, max\big(log(n_{ij} + 1)\big) \\ 0.175, \; for \; 0.35 \, max\big(log(n_{ij} + 1)\big) > n_{ij} \end{cases} \tag{6}$$

where $max\big(log(n_{ij} + 1)\big) = 3.67$ or $max(n_{ij}+1) = 4{,}657$ in this study. Finally, the spatial coverage of the grid cells is set

from (17.5° E; 30° N) to (40° E; 41° N), with a grid resolution of 0.1° x 0.1°. To take into account the uncertainties on back-trajectory path, a smoothing of attributed contributions was applied as recommended by Charron et al., 2000 using a smooth factor (i. e. the strength of a Gaussian filter) of 5.

## 3 Results

### 3.1 Air mass origin and meteorological conditions

In March 2015, weather conditions were characterized by varying hourly temperatures, ranging from 8 °C to 24 °C with an average of 13 °C ± 3 °C. The most favorable conditions for high levels of VOCs (high temperatures, clear skies and low winds) were observed from 8 to 10 March and from 25 to 27 March. Some rough-weather days, characterized by lower temperatures, heavy rain and strong winds also occurred (11 – 14 March, 20 – 22 March and 28 March) and rainy periods may participate to a larger development of vegetation. Throughout the campaign, the wind speed has shown a large degree of

variability, reaching up to 13 m.s$^{-1}$ (20 – 22 March) favoring the dispersion of pollutants. In the days with low winds favoring stable conditions, wind speed was below 2 m.s$^{-1}$. Due to the two mountains ranges, winds was blowing mainly from Western directions with wind speed up to 13 m.s$^{-1}$ (only observed under this wind sector) (Fig. 3).





Figure 2 shows the respective contribution of air mass origins observed during the campaign. The CAO station was mostly influenced by continental air masses originating from "Southwest Asia" (cluster 7 – 31 %), "Northwest Asia" (cluster 4 – 28 %), "West of Turkey" (cluster 5 – 10 %) and "Europe" (cluster 3 – 11 %) but also by marine air masses (cluster 2 – 14 %).  Note that, air masses categorized as "local" (cluster 0) occurred only on 23 and 24 March and may rather be considered as a transitory state between periods of air masses originating from Northwest Asia and West of Turkey. It is worth noting that March 2015 was characterized by an unusually high contribution of Southwest Asian air masses in the detriment of European air masses compare to the period 1997-2012 presented in Kleanthous et al., 2014. In this study, "local origins" will refer to contributions from Cyprus and "regionals origins" to air masses originating from the Mediterranean region. However, long-range transport contributions cannot be easily separated from local contributions.  Indeed, winds blowing from East and Southeast directions could lead to local influence from Larnaca and its surroundings and were observed when air masses originating from Europe, Northwest Asia and Southwest Asia (Sect. SI-3 in the supplement). Likewise, a second local anthropogenic influence may come from the Nicosia region corresponding to situations where the winds were observed in the Northeast direction and to air masses originating from Northwest Asia and Southwest Asia.

## 3.2 Atmospheric background conditions

Time series of gases (NO, $NO_2$, CO and $O_3$) and aerosol tracers ($PM_{2.5}$, $PM_{10}$ and BC split in $BC_{ff}$ and $BC_{wb}$ originating from fossil fuels and wood burning, respectively) are shown in Sect. SI-4 in the supplement.

NO levels remained low (below 0.3 ppb) during the whole campaign indicating a weak influence from directly surrounding sources. Likewise, the average $NO_2$ concentration was 0.9 ppb and peaks of concentration higher than 3 ppb were only met three times during the intensive field campaign (9 March, 18 March and 28 March). CO concentrations were in the range of 80 – 230 ppb with highest levels encountered on 10 March when the station received continental air masses originating from Southwest Asia. Levels of $O_3$ varied from 25 ppb to 60 ppb with the highest levels encountered on 9-10 March and 25-27 March. These periods correspond to the warmest days but also to air masses originating from Northwest and Southwest Asia, especially from the South of Turkey, an area marked by intense anthropogenic emissions of ozone precursors (see Sect. 3.5.2).

As $BC_{ff}$ was largely dominating (95% of BC) and it was suggested that BC was almost exclusively emitted from fossil fuel combustion. $BC_{wb}$ levels remained low during the whole campaign (< 0.2 µg.m$^{-3}$) indicating no significant wood burning event during the campaign. BC 30-min average levels increased (up to 2.2 µg.m$^{-3}$) when the station was under the influence of continental air masses (Northwest Asia and West of Turkey) corresponding to the warmest days of the period. Higher concentrations of $PM_{2.5}$ and $PM_{10}$ were noticed when the station received air masses originating from Southwest (concentrations up to 30 µg.m$^{-3}$ and 60 µg.m$^{-3}$, respectively). Among other influences, $PM_{2.5}$ and $PM_{10}$ concentrations were below 20 µg.m$^{-3}$ and 25 µg.m$^{-3}$, respectively.



## 3.3 VOCs mixing ratios

Twenty four different VOCs were measured on-line during the campaign. These compounds were classified into three major families: biogenic, anthropogenic and oxygenated VOCs (3, 15 and 6 measured VOCs, respectively). Biogenic compounds included only isoprene and monoterpenes, while anthropogenic compounds only included primary hydrocarbons (alkanes, alkenes, alkynes and aromatic compounds) almost exclusively emitted from human activities. As OVOCs come from biogenic and anthropogenic (primary and secondary) sources they have been presented separately. Statistical analysis, uncertainties and detection limits of the VOCs measurements are presented in Sect. SI-5 in the supplement.

Anthropogenic VOCs were the most abundant (49 % of the total concentration of the measured VOCs) and were mainly composed of $C_1$-$C_5$ compounds (76 % of the anthropogenic VOCs) but also of aromatics compounds (8 %). Biogenic compounds were mainly composed of monoterpenes (84 % of the total concentration of BVOCs) and had a small contribution to the total VOCs on average (about 5 %). With an average contribution of 46 % of the measured VOCs, OVOCs were mainly composed of methanol and acetone (47 % and 33 % of the total concentration of OVOCs, respectively).

An overview of VOCs concentrations observed at different background locations in the Mediterranean is presented in Table 3. Anthropogenic VOCs levels measured here were in the range of those measured at the rural/marine/remote sites of Cape Corsica (France), Finokalia (Greece) and Mesoregion (Greece) during fall and winter with the exception of ethane and propane that were higher at CAO. Concerning biogenic compounds, as for CAO, Cape Corsica and Finokalia sites are typical Mediterranean shrublands and isoprene concentrations measured at these sites during wintertime were of the same order of magnitude. Additionally, CAO seemed to present a higher contribution of monoterpenes in comparison with other comparable sites (e.g. Inia, Cape Corsica and Valderejo). OVOCs concentrations observed during the campaign were also in the same range as observed for the other Mediterranean sites but lower than summer campaigns values (e.g. Inia and Cape Corsica).

## 3.4 VOCs variabilities

### 3.4.1 Anthropogenic compounds

As expected for combustion tracers, CO co-varied well with acetylene, a long-lived anthropogenic species (Fig. 4). These two compounds exhibited higher levels when the station was influenced by air-masses imported from Northwest and Southwest Asia. Additionally, i-butane, a typical marker for evaporative sources, co-varied well with BC, another tracer of combustion processes suggesting they may be emitted by a common source point (e.g. traffic source including gasoline evaporation and exhaust emissions). Higher i-butane mixing ratios were observed when the site was under continental influence (from 6 March to 11 March) suggesting an additional contribution from more distant sources (regional origins). Short-lived compounds, propene and ethylene, co-varied suggesting a common local combustion source, even though a





tendency to increase was observed only for propene when the station was influenced by continental air masses (6-11 March) indicating a potential additional contribution from more distant sources.

### 3.4.2 Biogenic compounds

Results from two biogenic tracers, isoprene and α-pinene, are presented in Fig. 5. Their distinct diurnal pattern suggests that these compounds may not be emitted by the same biogenic source. The observed isoprene pattern followed the typical diel profile which depends on environmental parameters (temperature and solar radiation - Geron et al., 2000; Owen et al., 1997). Indeed, isoprene concentrations increased immediately at sunrise, indicative of local biogenic sources. Higher concentrations were also observed during the warmest days of the campaign (8-10 March) with a maximal temperature of 24°C. Isoprene presented high concentrations also during the nights of 8 March and 10 March. Surprisingly, α-pinene concentrations were elevated during nighttime than during the daytime. A possible interpretation could be that α-pinene was rapidly consumed by daytime oxidants due to its high reactivity. A similar nocturnal pattern has been observed elsewhere (Harrison et al., 2001; Kalabokas et al., 1997; Kalogridis et al., 2014) and was attributed to nocturnal emissions from monoterpenes storing plants from the understorey vegetation. These nocturnal maxima were enhanced by the shallow nocturnal boundary layer, and the low removal processes (i.e. low concentrations of oxidizing species) leading to higher concentrations.

### 3.4.2 Oxygenated compounds

With atmospheric lifetimes ranging from a few hours to several days, OVOCs can be emitted both from primary sources (mainly biogenic but also anthropogenic) and be produced by secondary processes related to the oxidation of anthropogenic and biogenic hydrocarbons. In Fig. 6, OVOCs present diurnal variabilities with large amplitudes when the temperature increased (9-12 March) suggesting a local biogenic contribution but also a potential secondary one (i.e. favorable conditions for photochemical processes). During the same period, the increase of background levels could be associated to a distant source of assumed anthropogenic origin in combination with high contribution of anthropogenic compounds of different lifetimes (see Sect. 3.4.1). Nocturnal variabilities were also observed, although not systematically, and were coinciding with an increase of temperature as observed during an intensive field campaign at Cape Corsica (Kalogridis, 2014). Possible interpretations would be linked to a dynamical phenomenon and a reinforcement of biogenic emissions.

### 3.5 Source apportionment of VOCs

In the following section, the 6-factor PMF solution (from simulations described in Sect. 2.3) is presented and discussed. Figure 7 shows the contributions of each factor to the species selected as input for the PMF model as well as the contribution of each species to the 6 factors determined by the PMF analysis. Figure 8 shows the time series of the 6 PMF factor contributions. In this section, lifetimes were estimated from kinetic rate constants of the reactions between the species of interest and OH (Atkinson and Arey, 2003) considering an average OH concentration of $1.0 \times 10^6$ molecules cm$^{-3}$ (Spivakovsky et al., 2000). Concerning isoprene and monoterpenes, we also considered their kinetic rate constants of their





reaction with ozone (Atkinson and Arey, 2003) with an average daily measured $O_3$ concentration of $1.7 \times 10^{12}$ molecules cm$^{-3}$ to estimate their lifetime.

VOCs result from direct emissions, chemistry, transport and mixing and therefore each individual factor cannot be attributed exclusively to one source category. Some of the computed factors may not be directly related to emission profiles but should rather be interpreted as aged profiles originating from different sources belonging to similar source categories (Sauvage et al., 2009). PMF was conducted to identify co-variation factors of VOCs that were representative of aged or local primary emissions as well as secondary photochemical transformations occurring during the transport of air masses sampled at this background site (Michoud et al., 2017).

### 3.5.1 Biogenic factors with distinct origins (factors 1 and 2)

Two biogenic factors were identified, "biogenic source 1" and "biogenic source 2". They are both composed of primary biogenic species but have shown distinct variabilities and origins justifying the division into two factors. Figure 9 shows the diurnal cycle of the biogenic factors along with meteorological parameters (temperature and solar radiation). CPF results of these factors are also investigated and are presented in Fig. 10.

The average contribution of the "biogenic source 1" factor is only 1.3 µg. m$^{-3}$ (7%) during the studied period due to its few episodic contributions. Indeed, from 8 March to 12 March, its relative contribution reaches 63 % of the total VOCs mass (Sect. SI-6 in the supplement) with peaks up to 34.8 µg. m$^{-3}$. This biogenic event occurred during the warmest period, with diurnal temperatures reaching up to 24 °C. The biogenic factor 1 explains 100 % of the variability of pinenes and the relative load of these species for this factor is 55 %. The lifetime of α-pinene and β-pinene in the troposphere was quite short (1.4 h and 2.8 h, respectively) indicating that these compounds were emitted exclusively by local vegetation. In addition, few OVOCs (methanol and acetone mainly) are also found in this profile. The contribution of these compounds to factor 1 is estimated at 21 %. These OVOCs are also known to be partly emitted from biogenic emission sources (Jacob et al., 2002; Schade et al., 2011; Schade and Goldstein, 2001; Seco et al., 2007; Singh et al., 2004). The diurnal profile of factor 1 exhibits higher contributions during nighttime, 18 h – 5 h LT (local time) (in agreement with the diurnal variability of α-pinene investigated in Sect. 3.4.2) and CPF analysis localizes this source in the South-Southwest direction from the sampling site. In that direction, the area is characterized as "high forest" (Fig. 10) corresponding to pines and/or oaks (Fall, 2012), known as high emitters of pinenes but also OVOCs (e.g. acetone, Janson and de Serves, 2001). Note that, South-Southwest wind directions were mainly encountered during nighttime (Sect. SI-3 in the supplement).

With an average contribution estimated at 6.1 µg. m$^{-3}$, this factor "biogenic source 2" is one of the two main VOCs factors observed during March 2015 at CAO, explaining 36% of TVOC. Factor 2 is mainly composed of OVOCs, such as methanol (77% explained), acetaldehyde (84% explained), acetone (50% explained) and MEK (59% explained). The total contribution of these oxygenated compounds to factor 2 is 79%, with more than 42% attributed to methanol. Most of these species are formed by the oxidation of both anthropogenic and biogenic compounds although some of them can also have a primary origin (e.g. MEK budget described in Yáñez-Serrano et al., 2016, and acetaldehyde budget in Millet et al., 2010;



Seco et al., 2007). Additionally, this factor profile includes isoprene (82% explained), a known chemical marker of primary biogenic emissions with a short lifetime (2.3 h), and its primary oxidation products (Spaulding et al., 2003), MACR and MVK (77% explained; lifetime of 9.6 and 13.8 h, respectively). Its diurnal profile exhibits a clear temperature and solar radiation dependency (Fig. 9), as already observed for isoprene in Sect. 3.4.2. Individually, oxygenated species have shown

the same daily variation as isoprene and its oxidation products confirming their direct link. In conclusion, factor 2 is also attributed to a biogenic source (mainly driven by isoprene emissions) while factor 1 is characterized by pinenes. Small amounts of $C_1$-$C_6$ alkanes are also found in this profile with a cumulated contribution to factor 2 of 12% and can be attributed to a mixing with other temperature-related sources or artefacts from the PMF model (Leuchner et al., 2015). CPF analysis of factor 2 localizes the corresponding biogenic emission area in the Northwest to the Northeast directions from the

sampling site with associated distinct land covers: "garrigues" and "other land" (Fig. 10) corresponding mainly to cultivated areas.

### 3.5.2 Anthropogenic factors (factors 3, 4 and 5)

Among the 6 identified PMF factors, three were attributed to anthropogenic sources (factors 3, 4, and 5) and were characterized by compounds of various lifetimes.

As highlighted from their co-variation observed using time series (Fig. 8), factors 4 and 5 correlated well (r of 0.59) from 6 to 12 March and from 26 to 29 March when the site was under the influence of air masses originating from Southwest Asia. During the same periods, these two factors were also in good agreement with factor 3 pattern. Individually, anthropogenic compounds investigated in Sect. 3.4.1 but also $C_6$-$C_{14}$ alkanes and fossil fuel combustion tracers (CO, $NO_2$ and BC) have shown similar behavior, higher mixing ratios and increase of their background levels specifically at that time.

These indications suggest that anthropogenic VOCs were potentially of the same origin when the station was influenced by air masses originating from Southwest Asia and this independently of their specific sources. The CF results, concerning factor contributions associated to Southwest Asian air masses alone (Fig. 11), seem to pinpoint the South coasts of Turkey as potential origin of this anthropogenic event observed at CAO. This region corresponds to densely populated areas of Turkey (including Adana and Gaziantep, with more than 1.6 million of inhabitants, the fifth and the sixth most densely populated

cities in Turkey, respectively) with expected high anthropogenic emissions due to intense industrial and maritime activities (e.g. the seaport of Mersin) and a dense road network. To study more specifically each of the three anthropogenic factors determined by PMF analysis, they will be investigated in the rest of this section excluding factor contributions associated to Southwest Asian air masses. Diel variabilities are shown in Fig. 12. CPF results were also investigated and are presented in Fig. 13.

Factor 3 only represents up to 4 % of the sum of measured VOCs. This factor is composed of primary anthropogenic VOCs, ethylene (100% explained) and toluene (14%), typical compounds of incomplete combustion processes with short-to-medium lifetime (1.4 d and 2 d, respectively), with a total contribution to this factor up to 55 %. Factor 3 displays faire correlation with ethylene (r = 0.94). Factor 3 seems to correlate with $NO_2$, CO and BC, which are



known to be relevant vehicle exhaust markers (r = 0.41, 0.40, 0.37, respectively). Even if the diurnal profile does not exhibit a clear variability except a slight increase in midmorning (Fig. 12), the time series shows a scattered variability (Fig. 8). CPF analysis of factor 3 does not clearly highlight a wind sector except maybe Southeast winds, supporting a local origin of this combustion source. The temporal variation was less pronounced on weekends than weekdays (average contribution of 0.5

5 $\mu g . m^{-3}$ and 0.7 $\mu g . m^{-3}$, respectively) consistent with human activities. This factor was hence attributed to "a short-lived combustion source" (Leuchner et al., 2015; Michoud et al., 2017).

The average contribution of factor 4 is approximately estimated at 2.7 $\mu g . m^{-3}$ (13 %) during the studied period. This factor is composed of medium-to-long-lived primary anthropogenic VOCs. The profile of this anthropogenic source exhibits a high contribution from alkanes, mostly i-/n-butanes with lifetimes ranging from 5 to 6 days and with more than 60

10 % of their variabilities explained by this factor, but also ethane (13 % explained – 47 days), propane (22 % - 11 days) and n-pentane (18 % - 3 days). The $C_4$-$C_5$ alkanes are found in the gasoline composition and evaporation source (Salameh et al., 2014, 2016). Ethane and propane can also appear as a significant profile signature of natural gas use. The total contribution of these compounds to factor 4 is up to 68 %. The diurnal variation of this source exhibits higher contributions during the day with peaks at 5 h - 6 h, 10 h, 12 h, 15 h LT corresponding to typical circulation of vehicles on the hill where the station

is located. CPF plot exhibits high contributions originating from North, Northeast and East wind sectors, e.g. in the direction of Nicosia region with its road traffic. Note that, more than 85 % of Cyprus passenger cars consume petrol (Eurostat, census 2013). Consequently, factor 4 can be viewed as "evaporation sources" including gasoline evaporation and with local origins. Regional origins were also not discarded as demonstrated in CF results (Fig. 11): potential emission areas could be the Southwest and Southeast coasts of Turkey.

Factor 5 only represents up to 5 % of the sum of measured VOCs with episodic contributions up to 27 % (Sect. SI-6 in the supplement). Factor 5 is composed of medium-lived primary anthropogenic species such as i-pentane (92 % explained by factor 5), n-pentane (69 %) and 2-methylpentane (27 %), with lifetimes ranging from 2 to 3 days and typically emitted by gasoline evaporation. The total contribution of these compounds for the factor 5 is 45 %. Significant contributions (34 % in total) for aromatic compounds, benzene (6 % explained by factor 5), toluene (48 %) and m107 (12 %), and some OVOCs,

methanol (4 %) and acetone (2 %), are also observed. The time series of factor 5 presented in Fig. 8 has shown that this factor was primarily episodic with high contribution which seemed to be originating from the South and Southeast wind sectors as suggested by the CPF plotted in Fig. 13. Regarding Cyprus emissions inventories (Papastavros et al., 2010), these directions may correspond to Larnaca region and different industrial implementations: several small industry areas are located in these directions but also industrial plants, e.g. three power plants (using heavy fuel and diesel for their boilers -

DLI, 2016) and two cement plants (only one is in operation in 2015 – DLI, 2016) located nearly South and Southeast coasts of Cyprus. A refinery was also referenced in Papastavros et al., 2010 but it was shutdown in 2004 and then converted into an oil storage terminal (all fuels consumed in the island are imported since then – DLI, 2016). Contribution of long-range transport cannot be discarded as well. The diurnal pattern of this factor (Fig. 12) has shown high contributions during nighttime, appearing to be relatively intense compared to other anthropogenic behaviors, and probably driven by dynamical



processes. Furthermore, diurnal profiles of factor 5 presented in Sect. SI-7 in the supplement displayed a distinction between contributions under South-Southeast wind directions and the other ones. In fact, the station was only under the influence of wind coming from the South and the Southeast directions from 18 h to 8 h LT explaining this nocturnal influence transported on the site by local wind circulation. The diurnal profile excluding high industrial contributions has shown a diurnal

variability (7 h – 18 h LT) in agreement with factor 4 (r= 0.64). This finding suggests that the diurnal contributions of factor 5 could be associated to evaporation sources including gasoline evaporation with local origins potentially associated to Nicosia region (Fig. 13) and/or regional origins, (South of Turkey - Fig. 11). Furthermore Salameh et al., 2016 found a "gasoline evaporation" factor, obtained from measurements realized in winter 2012 in the Eastern suburb of Beirut, with a profile resembling to factor 5 and high contributions corresponding to a particular wind direction corresponding to the

location of a fuel storage facility. Therefore, factor 5 may include evaporative sources and it was labelled as "industrial and evaporation sources" for that reason.

### 3.5.3 Regional background (factor 6)

With an average contribution estimated at 6.1 $\mu g \cdot m^{-3}$, factor 6 is the second main VOCs source observed during March 2015 at CAO explaining 36 % of TVOC. Factor 6 is mainly composed of long-lived primary anthropogenic VOCs, such as

ethane (70 % explained by this factor), propane (68 %), acetylene (74 %), and benzene (76 %) with lifetimes ranging from 11 to 47 days and typically emitted by natural gas use and combustion processes. The total contribution of these compounds was up to 70 % to factor 6, with more than 57 % only attributed to ethane and propane. Other anthropogenic NMHCs with shorter lifetimes compose this factor, such as toluene (33 %), 2-methylpentane (33 %) and $C_8$ aromatic compounds (27 %), although with a low contribution to factor 6 (below 3 %). Additionally, the profile of factor 6 explains to a large extent some

OVOCs such as acetone (38 %) and MEK (30 %). These species represent 19 % of the total load for this factor. Various sources, either primary or secondary (Singh et al., 2004; Yáñez-Serrano et al., 2016), can be associated to these oxygenated compounds characterized by long atmospheric lifetimes but their association with long-lived anthropogenic species suggested primary/secondary contributions. Indeed, several studies performed in urban or in rural areas (Goldstein and Schade, 2000; de Gouw et al., 2005; Legreid et al., 2007) have attributed about half of these oxygenated compounds

concentrations to the regional background pollution. Hence, the high abundance of long-lived species in combination with the lack of shorter-lived compounds suggests here aged air masses transported towards the sampling site. Furthermore, factor 6 exhibits variability similar to CO (see Fig. 14) supporting the identification of this factor as corresponding to long-lived compounds mainly of anthropogenic origin. The diurnal pattern of this factor (Fig. 15) is mainly based on the diurnal variation of ethane which is characterized by a nighttime maximum and a midday minimum. The diurnal profile of factor 6

displays higher levels during the night under a shallow inversion layer followed by a concentration decreased which could be attributed to the increase of the Planet Boundary Layer (PBL) and the vertical mixing. The potential source areas associated with this factor (Fig. 16) can be pinpointed to the South coasts of Turkey (Antalya, Mersin and Gaziantep provinces – Northwest Asian air masses). An additional minor continental contribution seems to be associated to the Peloponnese region




(Greece – European air masses). Finally, factor 6 can be interpreted as remote sources showing the continental regional background (Hellén et al., 2003; Leuchner et al., 2015; Sauvage et al., 2009). Hence, it is reported here as "regional background".

## 4 Discussions

### 4.1 Comparisons with VOCs measurements conducted in Mediterranean

#### 4.1.1 Comparisons with VOCs measurements conducted at a different background site of Cyprus

To the best of our knowledge, only one study has reported VOCs measurements performed at a background site of Cyprus with a similar typology than the CAO station. Indeed, during the CYPHEX summertime campaign (Cyprus Photochemical Experiment 2014), VOCs measurements were conducted mostly using a PTR-MS, at a rural site (Inia) located at the Southwest of the Troodos mountain range. Derstroff et al., 2016 have examined the temporal variation of VOCs observed during this campaign and investigated the impact of air masses originating from Eastern and Western Europe and of the marine boundary layer on VOCs, especially OVOCs.

Comparison between the two studies shows that each sampling site exhibited very low mixing ratios of primary anthropogenic compounds as expected from their remoteness from local pollution sources. For biogenic compounds, Derstroff et al., 2016 identified that the sparse local vegetation close to their site produced only modest concentrations of biogenic compounds (average concentration of ~0.26 $\mu$g. m$^{-3}$ and ~0.33 $\mu$g. m$^{-3}$ for isoprene and the sum of monoterpenes, respectively) with both daily maxima and a solar radiation dependent pattern. In our study, isoprene has shown also a daily variability and low concentrations (0.13 $\mu$g. m$^{-3}$) but consistent with the respective period of each campaign (CAO: winter/spring, Inia: summer), leading to different range of temperatures (average diel variations of the temperature between 11-15°C and 22-25°C for CAO and Inia, respectively). However, despite the lower temperature observed during our campaign, monoterpenes have shown higher levels (1.34 $\mu$g. m$^{-3}$) at CAO with nighttime maxima and potential production by oak and pine forests.

OVOCs observed during the CYPHEX campaign have shown high values (average concentration of ~4.08 $\mu$g. m$^{-3}$ and ~5.64 $\mu$g. m$^{-3}$ for methanol and acetone, respectively) and revealed little diel variations, indicating that local emission or production was minor in comparison to long-range transport. In this study, we found more pronounced OVOCs diel variations (not shown in this article but similar to biogenic source 2 diel variation), comparable average concentration of methanol (3.84 $\mu$g. m$^{-3}$) with mostly biogenic origins and lower concentration of acetone (2.72 $\mu$g. m$^{-3}$) with either biogenic origins (biogenic source 2) and primary/secondary anthropogenic origins (regional background). As a result, local origins were as significant as long-range transport in our study. Consistent with our findings, Derstroff et al., 2016 found higher OVOCs mixing ratios when air masses came from Eastern Europe and lower when arriving from Western areas.



### 4.1.2 Comparison with another PMF study performed at a remote site of the Mediterranean region

Michoud et al., 2017 have reported gas and aerosol measurements conducted at a French remote site of the Western Mediterranean region (Cape Corsica) during the ChArMEx SOP2 field campaign (summer 2013). They have performed a PMF analysis on a gas database made of 42 VOCs, including primary VOCs with anthropogenic and biogenic origins and

5 OVOCs and therefore offer a unique opportunity to provide a comprehensive characterization of VOCs for the entire Mediterranean. Note that, a same species had different lifetimes at each site since we considered a lower OH concentration, leading to higher lifetime, but consistent with respective measurement periods (CAO: winter/spring, Cape Corsica: summer).

On both sampling site, primary anthropogenic PMF factors were separated according to the lifetime of compounds which composed them suggesting a homogeneity phenomenon on the entire basin. Similarly to our regional background

factor, their "long-lived anthropogenic" factor was mainly composed of long-lived primary anthropogenic VOCs but also of oxygenated compounds (e.g. methanol and acetone), co-varying with CO and showing potential distant sources originated from Europe (the North of Italy and the Southeast of France) and Africa (Northeast of Tunisia). In this study, distant anthropogenic sources (regional background factor) have shown potential origins mainly associated to the Eastern Mediterranean region and represented a higher share of the sum of measured VOCs (CAO: 36 %, Cape Corsica: 16 %-17 %)

in the detriment of short-lived anthropogenic factor (CAO: 43 %, Cape Corsica: 21 %-23 %).

Different patterns were noticed concerning biogenic sources identified at these two Mediterranean receptor sites. We found a biogenic factor with a clear diurnal cycle correlated with temperature (biogenic source 2) but also a second biogenic source with nighttime maxima (biogenic source 1) and hence further distinguishing biogenic sources contributing to isoprene and monoterpenes concentrations contrary to Michoud et al., 2017. This difference with Michoud et al., 2017 may

be linked to the type of Mediterranean vegetation characterizing each measurement site or different processes could also occur. Biogenic sources represented significant contributors to the VOCs concentrations observed at these sampling sites (CAO: 36 %, Cape Corsica: 20%).

Finally, oxygenated compounds were incorporated in these two studies but they were apportioned differently. Our PMF analysis did not allowed to clearly separate primary sources from secondary sources contrary to Michoud et al., 2017

but we obtained a better distinction between anthropogenic and biogenic origins of measured OVOCs. Hence, OVOCs were mainly explained by biogenic sources in this study (64 %).

### 4.2 Exploring the potential of VOCs in secondary organic aerosols formation

### 4.2.1 Source apportionment of organic aerosols (OA)

Foremost, we only considered the organic fraction of NR-PM$_1$ in this study. Following the same method as Michoud et al.,

2017, OA concentration was apportioned into 3 factors using a constrained PMF analysis: HOA (Hydrogen-like OA), SV-OOA (Semi-Volatile Oxygen-like OA) and LV-OOA (Low-Volatile Oxygen-like OA). During our study, average mass contributions were 0.30, 1.14 and 1.89 µg. m$^{-3}$, for the 3 determined factors HOA, SV-OOA, and LV-OOA respectively and



for a total OA concentration of 3.33 µg. m$^{-3}$. Mass spectra profiles are presented in Sect. SI-8 in the supplement. Very low contributions of HOA were observed during the whole campaign, illustrating the weak influence of local anthropogenic sources of OA at the site, while secondary OA represented about 91 % of OA and was mainly composed of aged LV-OOA (63 % of OOA).

Figure 17 reports the time series of accumulated OA factors. From 4 March to 10 March, OA concentrations increased progressively up to 10 µg. m$^{-3}$ together with an increase of SV-OOA contributions to OA levels (up to 66 %.) Daily average SV-OOA concentration observed on 10 March was three times higher compared to 4 March. Similar patterns were also noticed for HOA and LV-OOA, but less pronounced (2 times higher). This period corresponded to a progressive increase of daily temperature (diel maximum temperatures increasing from 16 °C to 24 °C), leading to favorable conditions

for VOCs emission/formation, but also to air masses coming from Northwest and Southwest Asia, especially from the South of Turkey. OA concentrations were drastically reduced on 11 March due to precipitations and change of air masses origin. Other increases of OA concentrations were noticed, from 14-21 March and 22-29 March, consistent again with temperature trend but with less pronounced variability than the first period. From 26 March to 29 March, SV-OOA contribution to OA concentrations increased significantly when the station was under continental influence (cluster 7: Southwest Asia) whereas

the LV-OOA contribution was lower.

### 4.2.2 Relationship between VOCs and OA

Figure 17 aims at drawing a parallel between OA and VOCs to highlight the link between the two phases. Figure 18 shows accumulated contributions of OA and VOCs factors as a function of air masses origin. Note that, due to their local origins, biogenic VOCs factor contributions did not depend on long-range transport and higher contributions were only observed

when the station received air masses originating from local and Southwest Asia (clusters 0 and 7, respectively - Fig. 2) due to favorable days with the highest daily temperatures observed during March 2015.

We first investigated the average OA contributions according to air masses origin and we supported our findings based on the variability of VOCs factors. The lowest OA concentration was observed when the station was under the influence of marine air masses (cluster 2 - Fig. 2). Likewise, total VOCs concentration was also low during this period

mostly because of a low contribution of "regional background" VOCs factor that may be linked to dilution and removal processes and the fact that marine air masses have not been recently in contact with anthropogenic sources. The average OA contributions were twice higher when the station was under continental influence comparing to the one under marine influence. This is again consistent with the increasing trend of the VOCs regional background factor during this period. Furthermore, we noticed a higher continental influence on OA total contribution when air masses were originated from the

Eastern Mediterranean (clusters 4, 5 and 7 associated to air masses originating from West Asia – Fig. 2) compared to the Western Mediterranean (cluster 3 associated to air masses originating from Europe – Fig. 2). The stronger continental influence of air masses originating from the Eastern Mediterranean region has also led to a higher contribution of VOCs anthropogenic factors (Fig. 18) and higher organic aerosol emissions and formation.





We also investigated the variability of each OA factor individually using their diel variations (Sect. SI-9 in the supplement) and their variation according to air masses origin (Fig. 18). HOA diel variation has shown a slight increase during the day and then a second wave in the late afternoon. HOA was also correlated with VOCs anthropogenic factors (r= 0.39 and 0.40 for VOCs factors 4 and 5, respectively). Although HOA concentrations are very low, these results suggest a local origin (potentially Nicosia and Larnaca regions) but also an influence of long-range transport (particularly when the station received air masses originating from the Eastern Mediterranean region – Fig. 17 and Fig. 18).

It is interesting to note the weakly positive, but significant correlation found between freshly (locally) produced SV-OOA and VOCs biogenic sources (r= 0.53 and 0.47 for VOCs factors 1 and 2, respectively). Indeed, the diurnal variation of SV-OOA was characterized by a morning minimum followed by a daily maximum with an increase from 8 h to 12 h LT consistent with VOCs biogenic factor 2 variability (Sect. SI-9 in the supplement). Using time series of a selected day of the campaign (26 March – Fig. 19.b1), a possible link between SV-OOA and VOCs biogenic source 2 could be established based on their common daily variations. Moreover, during nighttime, SV-OOA variability exhibited a second maximum (Sect. SI-9 in the supplement) that could be linked with the biogenic influence from factor 1 and also with dynamical processes (higher contribution during the night under a shallow inversion layer). On 10 March (Fig. 19.b2), the increase of SV-OOA at 17 h LT was in phase with an increase of the VOCs biogenic source 1. In addition to these biogenic contributions, anthropogenic VOCs precursors may also contribute to SV-OOA formation. We found a good Pearson coefficient between SV-OOA and VOCs anthropogenic factor contributions (r=0.39 and 0.40 for VOCs factors 4 and 5, respectively) as illustrated on 28 March (Fig. 19.b3). Higher SV-OOA contributions were noticed when the station was under continental air masses than under marine air masses (SV-OOA from continental air masses was from 2 to 4 times higher than marine air masses) consistently with the respective variability of VOCs anthropogenic sources. In conclusion, the diurnal variability of SV-OOA seemed to be significantly influenced by biogenic contributions but also to a less extent by anthropogenic contributions.

Another interesting feature is the weakly positive, but significant correlation between LV-OOA and the VOCs regional background factor (r= 0.39), which is consistent with a highly processed (aged) regional background pollution advected to the receptor site. Both parameters have shown the same variability associated to air masses origin, with higher contributions under continental influence and slightly higher contributions when the air masses were originating from West Asia than Europe.

Finally, we have investigated more specifically VOCs and OA behaviors when the station was under the influence of air masses originating from the Southwest of Asia. SV-OOA contribution to OOA increased significantly under this influence (up to 86 % - Fig. 17).  As already discussed in Sect. 3.5.2, Southwest Asian air masses corresponds to an intense anthropogenic event potentially originated from the South coasts of Turkey leading to higher primary anthropogenic VOCs factor contributions (mostly VOCs factors 4 and 5). When the station received Southwest Asian air masses, each OA factor co-varied well with the sum of factors 4 and 5 contributions and this observation was confirmed estimating Pearson correlation coefficients (0.36 < r < 0.54). Even though potential anthropogenic emission areas originating from Northwest





of Asia were similar to those of Southwest Asian origin (Fig. 16), average OA concentration was lower for the Northwest Asia cluster. As stated before, the intense anthropogenic event observed when air masses coming from the Southwest of Asia also occurred during an intense biogenic influence period. Indeed, when the station received air masses categorized in Southwest Asia cluster, both OOA factors contributions were well correlated with VOCs biogenic source 2 contributions (r > 0.48) and only SV-OOA factor was well correlated with VOCs biogenic source 1 (r=0.58). The most evident relationship between biogenic VOCs and OA was observed on 10 March (Fig. 18.b2) and showed an increase of SV-OOA concentration of 4 $\mu g. \, m^{-3}$ in few hours coinciding with the maximal contribution of VOCs biogenic source 1 (up to 34.8 $\mu g. \, m^{-3}$) while no significant anthropogenic contributions could be observed at that period.

## 5 Conclusion

At the crossroads of intense local and long-range transported natural and human-made emissions, the Eastern Mediterranean is characterized as a sensitive region strongly affected by air pollution. This pollution is partly due to the combination of human pressure concentrated in coastal cities with more distant sources transported from 3 continents. This complex mixture of air pollutants will have impacts on human health, ecosystems and climate. However, observational data collected in the Eastern Mediterranean remain scarce, preventing any comprehensive assessment the various impacts of this pollution in the region.

Within the framework of ChArMEx and ENVI-Med "CyAr" programs, a 1-month intensive field campaign held in March 2015 at a background site of Cyprus, has provided an unique opportunity to provide insights in the various origins and fates of VOCs in the Eastern Mediterranean region. Real-time measurements of a large number of VOCs, including alkanes, alkenes, alkyne, aromatics compounds and oxygenated VOCs, have been performed allowing the evaluation of VOCs concentration levels in ambient air, improving our understanding of their major sources in the area and describing their variabilities and their potential origins. This period of the year typically offers contrasting conditions in terms of air masses transport (geographical origin, fast/low transport) and weather (temperature, humidity, precipitations ...). The remote location of the measurement site offered ideal experimental conditions to monitor long-range transported clean/polluted air masses from 3 continents (Europe, Africa and Asia) with limiting influence of local anthropogenic areas. The site is also surrounded by widespread vegetation with intense local biogenic emissions making possible the investigation of biogenic and anthropogenic interactions on air mass composition.

Low levels of short-/medium-lived primary anthropogenic VOCs (e.g. average concentration of 0.35 and 0.37 $\mu g. \, m^{-3}$ for ethylene and benzene, respectively) were measured confirming the background typology of the site. Anthropogenic VOCs exhibited higher background levels when the station was under the influence of distant sources imported by continental air masses mainly originating from West Asia. Additionally, significant levels of short-lived biogenic VOCs (up to 0.62 and 11.63 $\mu g. \, m^{-3}$ for isoprene and α-pinene, respectively) were observed with distinct diurnal pattern (daily and nighttime maximum, for isoprene and α-pinene, respectively). Elevated mixing ratios of OVOCs (e.g. up



to and 12.82 and 6.46 $\mu g \cdot m^{-3}$ for methanol and acetone) were largely dominating the VOCs budget and had both biogenic and anthropogenic origins in variable proportions. OVOCs can be emitted from primary sources (mainly biogenic) and can be produced by secondary sources related to the oxidation of anthropogenic and biogenic hydrocarbons making more difficult to directly assess their origins.

A source apportionment (PMF) analysis was conducted to better identify co-variation factors of VOCs that were representative of aged or local primary emissions as well as secondary photochemical transformations occurring during the transport of air masses. The US EPA PMF v. 5.0 was applied to the dataset composed by 20 VOCs of anthropogenic and biogenic origins and a total of 1,179 atmospheric data points with a 30min-time resolution. The best PMF solution allowed the decomposition of VOCs into six different sources; two factors attributed to biogenic sources (biogenic sources 1 and 2),

three anthropogenic factors (short-lived combustion sources, evaporative sources, industrial and evaporative sources) and a last factor associated to regional background pollution. The biogenic sources and the regional background were found to be the largest contributors to the VOCs concentrations observed at our background site, with a total contribution of 79 %. To investigate variability and origin of these factors, we based our observations on diel variabilities, CPF and CF plots. Factors attributed to biogenic sources 1 and 2, driven by pinenes and isoprene/OVOCs emissions respectively, have shown

contrasted diurnal profiles (nighttime vs. daily maxima) and distinct origins (oak and pine forests vs. garrigues). Factors attributed to anthropogenic sources were characterized by compounds of various lifetimes and were influenced by several geographic locations, from local origins to distant potential emission areas such as the South coasts of Turkey. The last factor was characterized by long-lived primary anthropogenic VOCs and OVOCs and was co-varied with CO supporting the identification of this factor as continental regional background.

Benefiting from real-time organic aerosol measurements performed by ACSM technique, a parallel between organic aerosol and gas phase composition was conducted during our study to better highlight the relationship between the two phases. Organic aerosol (NR-PM$_1$ OA) measurements were apportioned into 3 factors, HOA, SV-OOA and LV-OOA. Very low contributions of HOA were observed during the whole campaign, illustrating the weak influence of locally emitted anthropogenic sources of OA at the site while secondary OA represented about 88 % of OA with aged OOA contributing for

approximately 56 % of this secondary OA fraction. The diurnal variability of SV-OOA seemed to be influenced by local biogenic contributions while LV-OOA seemed to be of regional background origin. As expected, air mass origin revealed lower contributions of OA and VOCs when clean marine air masses were advected to the station. OA and anthropogenic sources of VOCs were higher when the station was under continental influence coming from East Mediterranean region (West Asian air masses). The highest total concentration of OA was observed when the station was under the influence of air

masses coming from the Southwest of Asia. The intense anthropogenic event potentially associated to the South coasts of Turkey transported on the site air masses rich in semi-volatile OA (less oxidized aerosols) associated with an increase of anthropogenic VOCs sources contribution. This event also occurred during a period with intense biogenic influence leading to an increase of biogenic VOCs sources contribution due to favorable temperatures. The organic fraction of aerosols during this period most probably resulted in a combination of both biogenic and anthropogenic sources in variable proportions.



In perspective of this study, we are writing a further paper dedicated to local biogenic emissions impacting the atmospheric composition of the site and the chemistry associated as we noticed they were significant contributors of SOA in this study. This second paper will also be a good opportunity to further compare biogenic compounds measured at CAO and Cape Corsica (levels, speciation, variability and processes). Additionally to this campaign, an annual campaign has been conducted. Twenty primary (anthropogenic/biogenic) VOCs have been measured continuously in order to study seasonal variabilities.

**Acknowledgments:** This study was supported by ChArMEx, ENVI-MED and ACTRIS-2 (European Union's Horizon 2020 research and innovation programme, grant agreement No 654109), CEA and CNRS. The authors would like to thank N. Mihalopoulos for his help in the establishment of the CAO observatory; F. Dulac and E. Hamonou for managing with enthusiasm the ChArMEx project.



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





**Table 1: Details of technics and measurements of VOCs from 01 March 2015 to 29 March 2015.**

[a] **ethane, ethylene, propene, propane, i-butane, n-butane, acetylene, i-pentane and n-pentane**

[b] **2-methylpentane, benzene, toluene, ethylbenzene, m,p-xylenes, o-xylene, α-pinene and β-pinene**

[c] **m33 (methanol), m42 (acetonitrile), m45 (acetaldehyde), m59 (acetone), m69 (isoprene), m71 (MVK+MACR and eventually Isoprene Hydroxy Hydroperoxide (ISOPOOH)), m73 (MEK), m79 (benzene), m93 (toluene), m107 (xylenes + $C_7$-species) and m137 (monoterpenes)**

| Instrument | GC-FID Chromatrap | GC-FID AirmoVOC C6-C12 | PTR-QMS Scan mode (33 amu – 137 amu) |
|---|---|---|---|
| Time Resolution (min) | 30 | 30 | 10 |
| Number of samples | 1282 | 1321 | 3879 |
| Temporal coverage (%) | 94 | 97 | 93 |
| Detection limit ($\mu g \cdot m^{-3}$) | 0.02 - 0.19 | 0.03 - 0.09 | 0.01 - 0.08 |
| Uncertainties $\frac{U(X)}{X}$ [min, mean, max] (%) | [14, 39, 73] | [18, 36, 53] | [18, 22, 44] |
| Species | 9 $C_2$ - $C_5$[a] VOCs | 9 $C_6$ - $C_{10}$[b] VOCs | 11 mass[c] |
| Reference | Gros et al., 2011 | Xiang et al., 2012 | Blake et al., 2009; de Gouw and Warneke, 2007; Taipale et al., 2008 |





**Table 2: Input information and mathematical diagnostic for the results of PMF analysis.**

| Input information | | |
|---|---|---|
| Samples | N | 1179 |
| Temporal coverage | | 87% |
| Species | M | 20 |
| Factors | P | 6 |
| Runs | | 100 |
| Nb. Species indicated as weak | | 2 |
| $F_{peak}$ | | 0.8 |
| **Model quality** | | |
| Q robust | Q(r) | 14130.3 |
| Q true | Q(t) | 14397.5 |
| Q(t)/Q expected | | 1.03 |
| Maximum individual standard deviation | IM | 0.20 |
| Maximum individual column mean | IS | 1.45 |
| Mean ratio (modelled vs. measured) | Slope(TVOC) | 1.00 |
| $TVOC_{modelled}$ vs. $TVOC_{mesured}$ | R(TVOC) | 0.98 |
| Nb. of species with $R^2 > 0.6$ | | 14 |
| Nb. of species with $1 > slope > 0.6$ | | 14 |



**Table 3: Comparison of mean concentrations of selected VOCs with ambient levels observed in the literature in the Mediterranean region from different atmosphere. Concentrations are expressed in µg.m$^{-3}$.**

[1] **VOCs measured by PTR-MS**

[a] **this study, [b] Derstroff et al., 2016, [c] ChArMEx database, [d] Navazo et al., 2008, [e] Lo Vullo et al., 2016, [f] Liakakou et al., 2009, [g]**
5 **Moschonas and Glavas, 2000, [h] Michoud et al., 2017; Kalogridis, 2014, [i] Seco et al., 2011, [j] Davison et al., 2009.**

| NMHCs | Cyprus Atmospheric Observatory[a] Background Winter 2015 | Ineia (Cyprus)[b] Summer 2014 | Cape Corsica (France)[c] Background Winter 2013-2014 | Valderejo (Spain)[d] Rural 2003-2004 | Mt. Cimone (Italy)[e] Remote Winter 2010-2014 | Finokalia (Greece)[f] Marine Winter 2004-2006 | Messorougion (Greece)[g] Mountain Fall 1996 |
|---|---|---|---|---|---|---|---|
| **Ethane** | 3.05 | | 2.88 | 1.80 | | 3.05 | |
| **Ethylene** | 0.35 | | 0.49 | 0.28 | | | |
| **Propane** | 2.20 | | 1.77 | 0.86 | 1.16 | 1.39 | 1.75 |
| **Propene** | 0.19 | | 0.08 | 0.16 | | | 0.11 |
| **i-Butane** | 0.32 | | 0.41 | 0.29 | 0.29 | 1.32 | 0.17 |
| **n-Butane** | 0.54 | | 0.70 | 0.48 | 0.54 | 1.36 | 0.77 |
| **Acetylene** | 0.71 | | 0.56 | 0.42 | 0.45 | | 1.60 |
| **i-Pentane** | 0.25 | | 0.38 | 0.27 | 0.22 | 0.66 | 0.33 |
| **n-Pentane** | 0.21 | | 0.30 | 0.18 | 0.18 | 0.63 | 0.21 |
| **Benzene** | 0.37 | 0.08-0.12 | 0.53 | 0.24 | 0.44 | 0.88 | 0.55 |
| **Toluene** | 0.19 | 0.07 | 0.41 | 0.38 | 0.22 | 0.69 | 0.50 |

| BVOCs | Cyprus Atmospheric Observatory[a] Background Winter 2015 | Ineia (Cyprus)[b] Summer 2014 | Cape Corsica (France)[c] Background Winter 2013-2014 | Cape Corsica (France)[h] Background Summer 2013 | Valderejo (Spain)[d] Rural 2003-2004 | Montseny (Spain)[i] Rural Spring 2009 | Finokalia (Greece)[f] Marine Winter 2004-2006 |
|---|---|---|---|---|---|---|---|
| **Isoprene** | 0.13 | 0.22-0.29 | 0.07 | 0.55 | 0.14 | 0.05-0.20 | 0.51 |
| **α-Pinene** | 0.33 | | | 0.61 | | | |
| **β-Pinene** | 0.34 | | | 0.80 | | | |
| **Monoterpenes** | 1.34 | 0.28-0.37 | | 2.30 | 0.57 | 0.03-0.38 | |

| OVOCs[1] | Cyprus Atmospheric Observatory[a] Background Winter 2015 | Ineia (Cyprus)[b] Summer 2014 | Cape Corsica (France)[h] Background Summer 2013 | Montseny (Spain)[i] Rural Spring 2009 | Castel-porziano (Portugal)[j] Rural Spring 2007 |
|---|---|---|---|---|---|
| **Methanol** | 3.84 | 3.43-4.73 | 4.20 | 1.71-3.61 | 4.39 |
| **Acetaldehyde** | 0.83 | 0.45-0.68 | 0.60 | 0.42-1.21 | 1.80 |
| **Acetone** | 2.72 | 4.83-6.44 | 6.20 | 1.90-3,74 | 3.82 |
| **MVK+MACR** | 0.09 | 0.12-0.13 | 0.27 | 0.03-0.14 | |





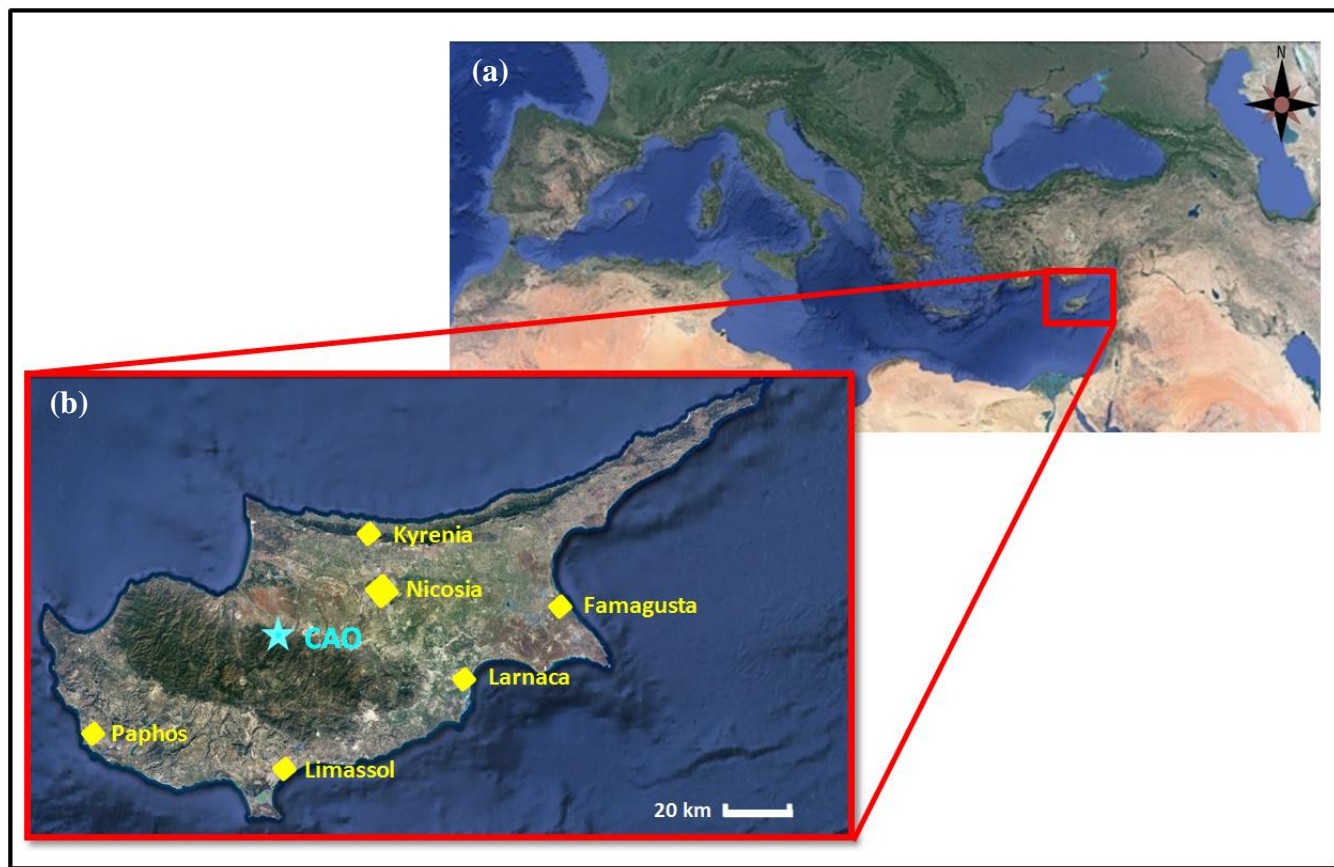

**Figure 1: Maps of Cyprus and Mediterranean region. (a) - Position of Cyprus in the Mediterranean region. (b) – The sampling site and major Cypriot agglomerations are displayed as blue star and yellow diamonds, respectively.**



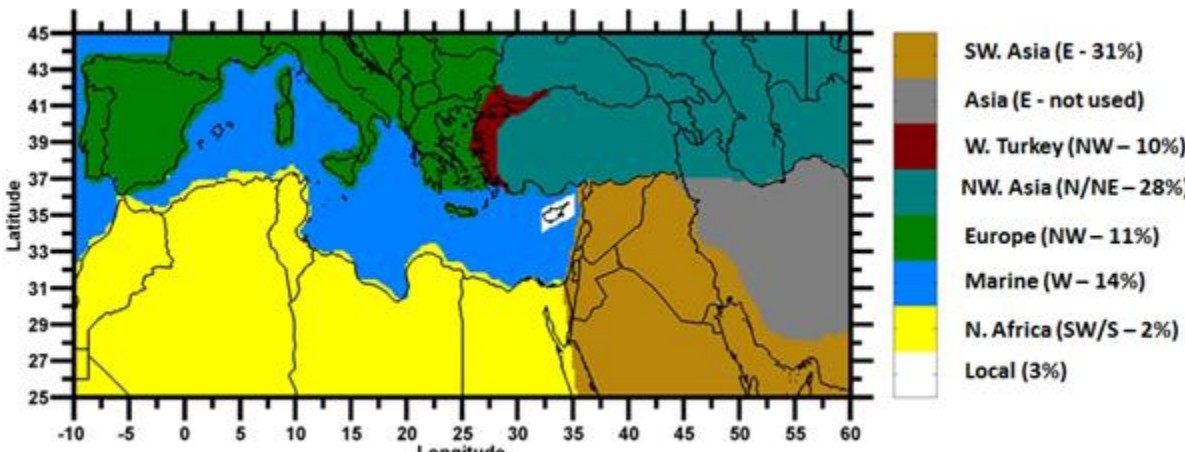

**Figure 2: Classification of air masses which impacted the site during the intensive field campaign of March 2015 and their relative contribution. A fraction of 2 % (not shown here) is attributed to air masses of mixed origins.**





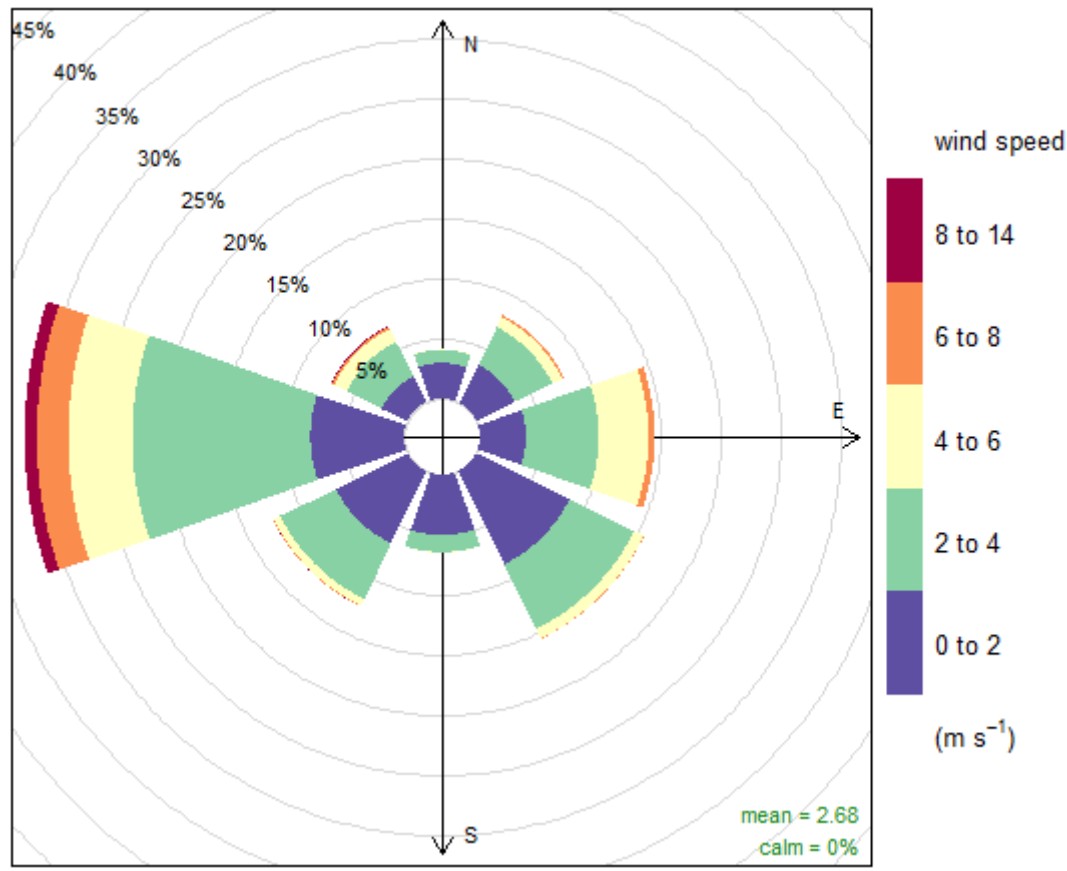

**Figure 3: Wind rose of March 2015. Contribution expressed in % corresponds to the frequency of counts by wind sector.**





**Figure 4: Time series of a selection of anthropogenic VOCs (acetylene, i-butane, ethylene and propose – blue lines) in comparison with air masses origin, meteorological parameters (temperature displayed as red lines) and air combustion tracers (CO and BC – black lines).**



**Figure 5: Time series of a selection of anthropogenic VOCs (isoprene and α-pinene – green lines) in comparison with meteorological parameters (solar radiation, temperature and relative humidity; displayed as orange, red and blue lines, respectively).**





**Figure 6: Time series of a selection of OVOCs (methanol, acetaldehyde, acetone, purple lines), temperature (red line) and air mass origin clustering.**





**Figure 7: Chemical profiles of the 6-factor PMF solution (20 VOCs). The contribution of the factor to each specie (µg. m⁻³) and the percent of each specie apportioned to the factor are displayed as a grey bar and a color circle, respectively. Factor 1 - biogenic source 1; factor 2 - biogenic source 2; factor 3 – short-lived combustion source; factor 4 – evaporative sources; factor 5 – industrial and evaporative sources; factor 6 – regional background.**





**Figure 8: Time series of VOCs factor contributions ($\mu g.\,m^{-3}$). Factor 1 - biogenic source 1; factor 2 - biogenic source 2; factor 3 – short-lived combustion source; factor 4 – evaporative sources; factor 5 – industrial and evaporative sources; factor 6 – regional background.**









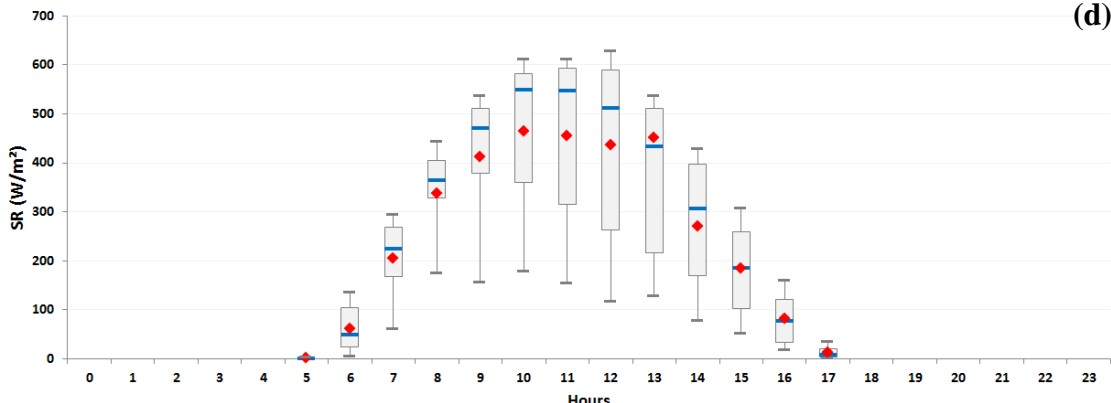

(d)

**Figure 9: Diel variation of the VOCs factor contribution (A and B), temperature and solar radiation (C and D) represented by hourly box plots. Blue solid line represents the median value, the red marker represents the mean value and the box shows the InterQuartile Range (IQR). The bottom and the top of box depict the first and the third quartiles (i. e. Q1 and Q3). The ends of the whiskers correspond to first and the ninth deciles (i. e. D1 and D9). Time is given as local time.**



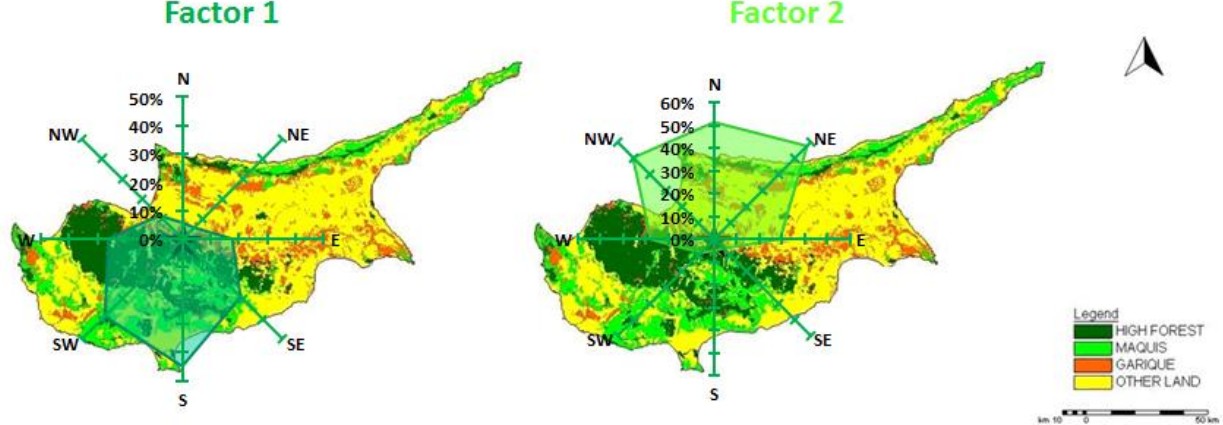

**Figure 10: Rose of the CPF for VOCs factors 1 & 2 in function of vegetation. Credit picture: Vegetation map of Cyprus (Natural resource information and remote sensing center 1998).**







**Figure 11: Potential source areas contributing to the 3 anthropogenic VOCs factors, determined using the CF model 5-days back-trajectories from HYSPLIT model, as a function of air masses origin. Contributions are in µg.m$^{-3}$. Cluster 0 – Local; Cluster 1 – N. Africa; Cluster 2 – marine air masses; Cluster 3 – Europe; Cluster 4 – NW Asia; Cluster 5 – West of Turkey; Cluster 7 – SW Asia.**



**Figure 12: Diel variation of the factor contributions (A, B and C) represented by hourly box plots. Diurnal profiles do not include contributions obtained when the site was under the influence of air masses categorized in cluster 7. Blue solid line represents the median contribution, the red marker represents the mean contribution and the box shows the InterQuartile Range (IQR). The bottom and the top of box depict the first and the third quartiles (i. e. Q1 and Q3). The ends of the whiskers correspond to first and the ninth deciles (i. e. D1 and D9). Time is given as local time.**





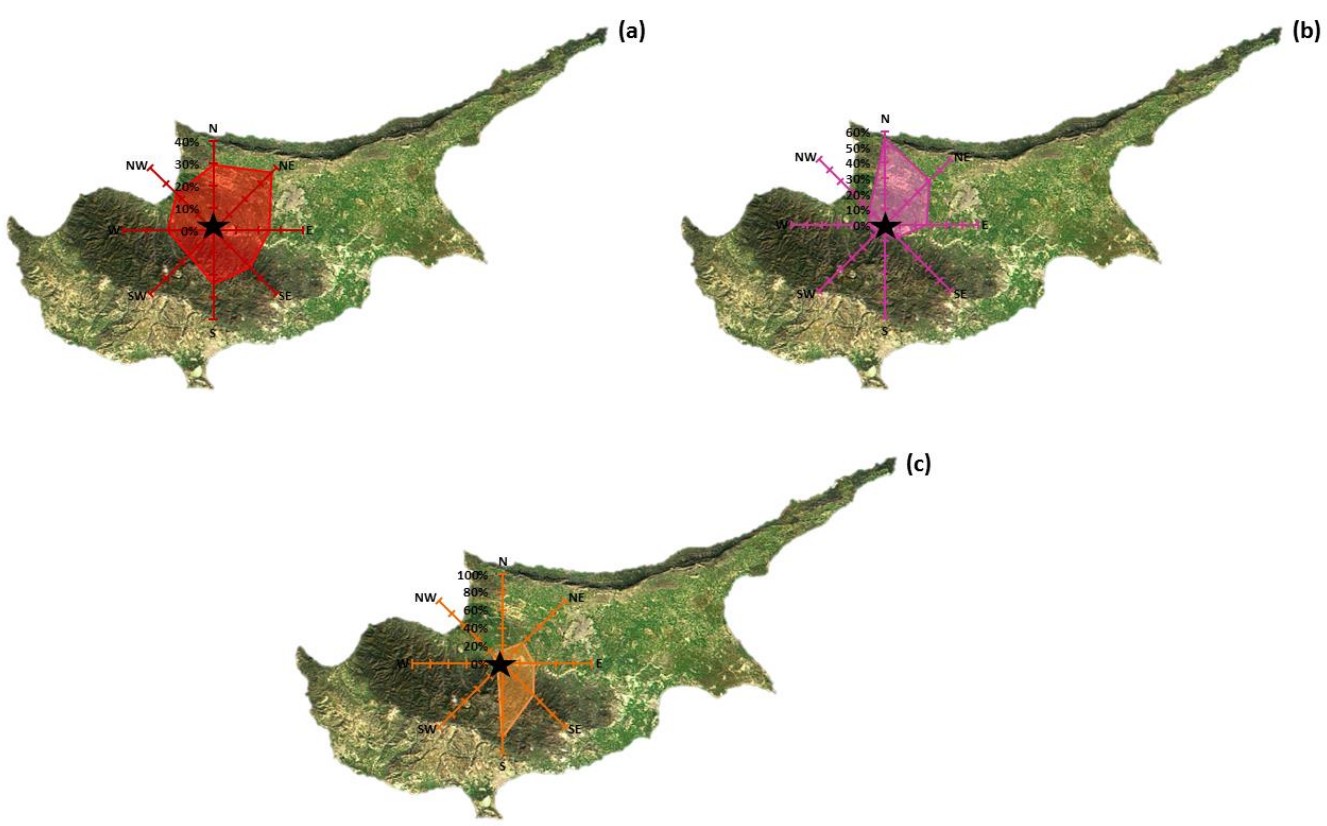

**Figure 13: Rose of the CPF for VOCs factors 3, 4, 5 (A, B, C, respectively). CPF results do not include contributions obtained when the site was under the influence of air masses categorized in cluster 7.**





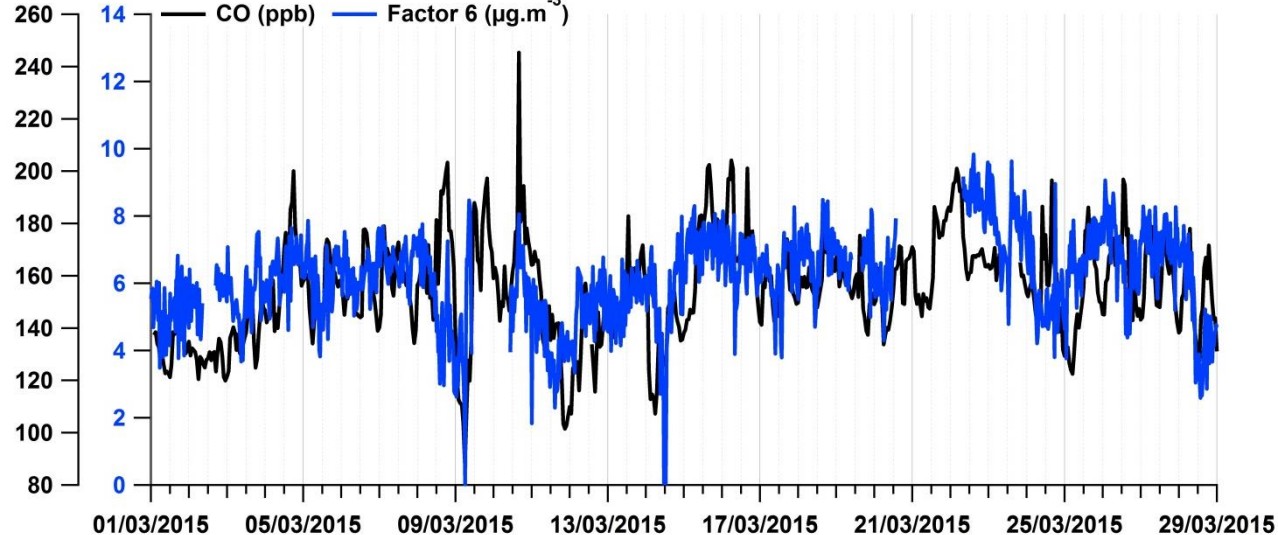

**Figure 14: Time series of the VOCs factor 6 contribution (blue line) and CO (dark line).**





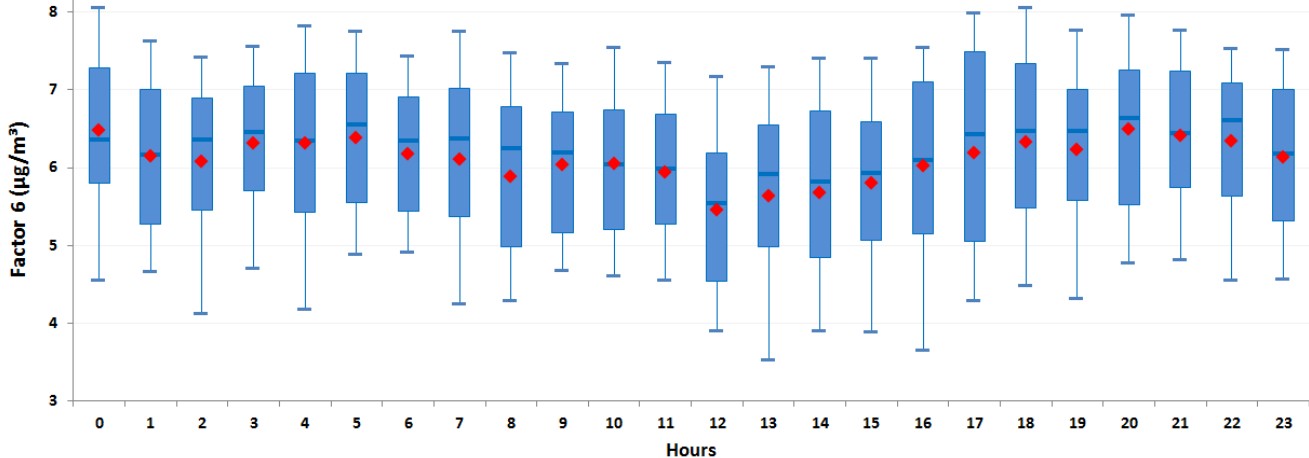

**Figure 15: Diel variation of the VOCs factor 6 contributions represented by hourly box plots. Blue solid line represents the median contribution, the red marker represents the mean contribution and the box shows the InterQuartile Range (IQR). The bottom and the top of box depict the first and the third quartiles (i. e. Q1 and Q3). The ends of the whiskers correspond to first and the ninth deciles (i. e. D1 and D9). Time is given as local time.**





**Figure 16: Potential source areas contributing to the VOCs factor 6 in function of air mass origins. Contributions are in units of μg.m⁻³. All – without distinction of air mass origins; C2 – marine air masses; C3 – Europe; C4 – NW Asia; C5 – West of Turkey; C7 – SW Asia. Low numbers of samples associated to clusters 0 and 1 (Local and N. Africa, respectively – figure 2) don't allow to apply CPF analysis only considering these air masses origin.**



**Figure 17: Accumulated time series of OA and VOCs factor contribution, time series of temperature and air masses origin (C: cluster). VOCs factors: factor 1 - biogenic source 1; factor 2 - biogenic source 2; factor 3 – short-lived combustion source; factor 4 – evaporative sources; factor 5 – industrial and evaporative sources; factor 6 – regional background. OA factors: HOA - hydrogen-like OA; SV-OOA – semi-volatile oxygen-like OA; LV-OOA – low-volatile oxygen-like OA.**





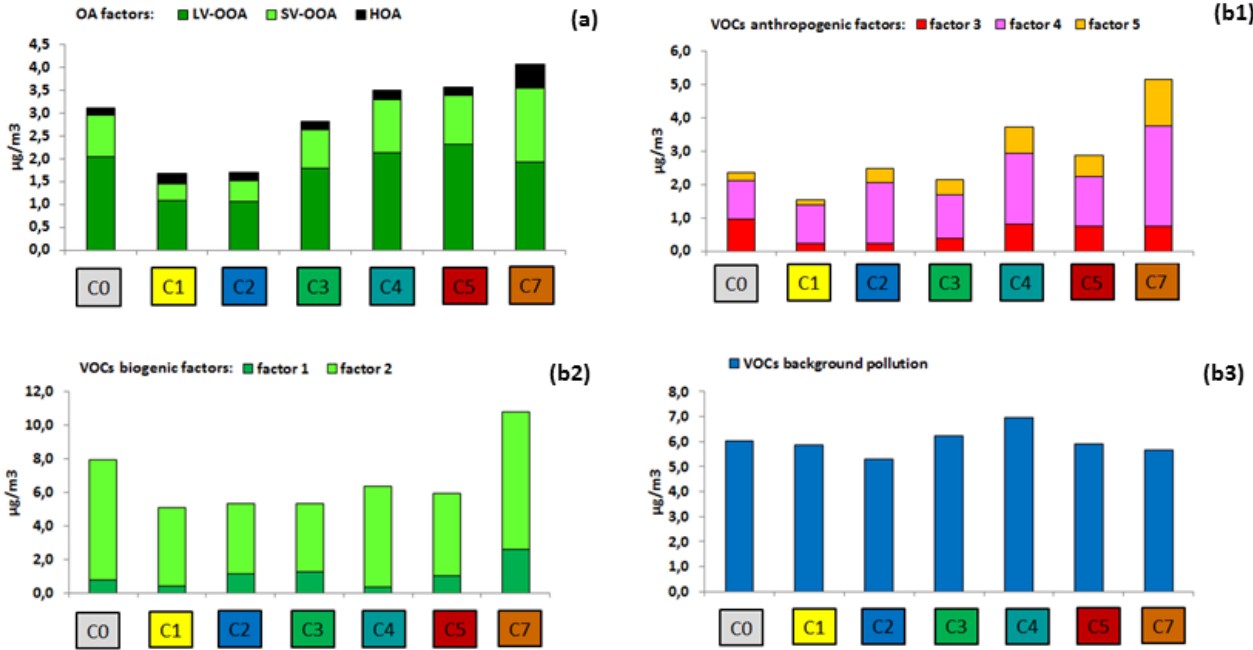

**Figure 18: Accumulated average contributions of the OA and VOCs factors (figures a and b, respectively) in function of air mass origins. Classification of air masses: C0 - Local; C1 - N. Africa; C2 - Marine; C3 - Europe; C4 - NW. Asia; C5 - W. of Turkey and C7 - SW. Asia. VOCs factors: factor 1 - biogenic source 1; factor 2 - biogenic source 2; factor 3 – short-lived combustion source; factor 4 – evaporative sources; factor 5 – industrial and evaporative sources; factor 6 – regional background. OA factors: HOA - hydrogen-like OA; SV-OOA – semi-volatile oxygen-like OA; LV-OOA – low-volatile oxygen-like OA.**









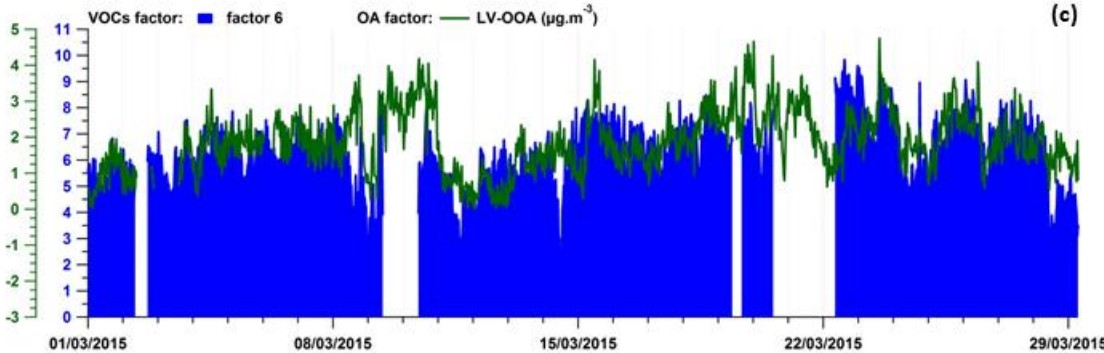

**Figure 19: Time series of OA and VOCs factor contributions of March 2015 (figures A and C) or of selected days of the intensive field campaign (figures B1, B2, B3) to point out co-variations and confirm average daily variations presented in figure S9 in the supplement. VOCs factors: factor 1 - biogenic source 1; factor 2 - biogenic source 2; factor 3 – short-lived combustion source; factor 4 – evaporative sources; factor 5 – industrial and evaporative sources; factor 6 – regional background. OA factors: HOA - hydrogen-like OA; SV-OOA – semi-volatile oxygen-like OA; LV-OOA – low-volatile oxygen-like OA.**