# Peer review of "Origin and variability of volatile organic compounds observed at an Eastern Mediterranean background site (Cyprus)"

_Atmospheric Chemistry and Physics, 2016_

## Referee Comment (RC1) · Anonymous Referee #1 · 18 May 2017

General comments:

The article presents an extremely thorough and comprehensive overview of the field site on the Island of Cyprus. It details the analytical methods and instruments used; describes the chemical and broad aerosol composition; identifies the most likely sources of the pollutants observed; and describes the observations in the broader context of the "Mediterranean region". It provides a very good description of the site for anyone wishing to make observations in this region. The article, however, is (inevitably) very long and I fear that some of the impact of the paper may be lost due to its length. That said, this is no reason to exclude the paper from publication and so I recommend that it be published subject to the authors addressing the following comments.

[Figure]

Specific comments:

A wide range of VOCs are reported from the site and yet 1,3-butadiene a commonly reported anthropogenic compound which fits within the range of VOCs covered by the instruments used (and has important health and atmospheric chemistry impacts) does not appear to be reported. It would likely be of interest to many others in the field. Could this be included or were problems encountered for measurement of this compound?

Some of the VOCs reported here are unsaturated compounds which are known to be highly reactive to ozone. There are known potential interferences caused when measuring these compounds in the atmosphere with GC systems. Were any measures taken to remove the ozone from the samples? If not, were the additional uncertainties resulting from ozone reactions considered here? Perhaps inclusion of a short discussion of these issues could be added?

The PMF analyses show a couple of gaps in the time-series around the 10th and 21st March (Figures 8, 14, 17 and 19). I may have missed it in the text, but the authors should confirm that this due to a loss of data (assuming it is) rather than being unable to assign these periods to any of the 6 factors identified.

Some of the differences described when comparing the PMF analyses from different site in the region appear to be (possibly) explained by differences in how the compounds were apportioned. Early in the manuscript the authors describe how (paraphrasing) the PMF analysis was tuned to give optimal results for this site over this measurement period and presumably Michoud et al. have done similar for that study. Perhaps the only real way to make a valid comparison between the PMF results for the region would be to merge/re-assign the PMF factors to fit both sites? I can see that this may not be suitable for local pollution events, but would perhaps better describe the main factors affecting the "Mediterranean region". I don't expect the authors to actually do this here, but they could perhaps summarise the possible benefits and drawbacks of such a study.

Technical comments/corrections:

Page 2, line 29: "A robust tool to identify emission sources is Positive Matrix Factorization (PMF - Paatero, 1997; Paatero and Tapper, 1994) is one of the various tools developed to identify emission sources. Over the last decade..."

Should read something like: "A robust tool to identify emission sources is Positive Matrix Factorization (PMF - Paatero, 1997; Paatero and Tapper, 1994) . Over the last decade..."

Page 3, line 22: "...and especially its evolution..."

Should read: "...and especially their evolution..."

Page 3, line 24: "Affected by important pollution sources, the Mediterranean is a sensitive region affected by both particulate and gaseous pollutants."

Should read something like: "The Mediterranean is a sensitive region affected by both particulate and gaseous pollutants."

Page 4, line 17: "...(geographical origin, fast/low transport)..."

Should read: "...(geographical origin, fast/slow transport)..."

Page 4, line 19: "It is ideally located, close from the coast"

Should read: "It is ideally located, close to the coast"

Page 4, line 23: "Our study performed in the Eastern Mediterranean will therefore offer a unique opportunity to provide a comprehensive characterization of VOCs for the entire Mediterranean."

This is a very grand statement and, I would suggest, is overstating the robustness of the study somewhat. Perhaps "detailed" rather than "comprehensive" would be more accurate?

Page 4, line 30: "...performed at an another background site of..."

Should read: "…performed at another background site of…"

Page 5, line 12: "…European Research Infrastructure fir the observation of…"

Should read: "…European Research Infrastructure for the observation of…"

Page 5, line 17: "…more than 35 km of main Cypriot agglomerations, with very poor influences of these anthropogenic emission areas.".

Do the authors mean "limited influences"?

Page 6, line 2: "AirmoVOC C6-C12, the measurement of C6-C10 hydrocarbons"

Why were measurements not made up to C12? Perhaps a statement could b emade in the text for clarity here?

Page 13, line 21: "The most favorable conditions for high levels of VOCs (high temperatures, clear skies and low winds) were…"

I disagree with this statement as it is written. Warmer temperatures may lead to greater emissions from biogenic and also evaporative sources, however emissions from heating sources etc are normally observed to decrease. Warmer conditions are also commonly associated with greater boundary layer height leading to lower concentrations. Can the authors re-phrase this for improved clarity?

Page 13, line 62: "…and rainy periods may participate to a larger development of vegetation."

I suggest changing "participate" to "contribute"

Page 16, line 10: "…elevated during nighttime than during the daytime."

Should read: "…elevated during nighttime compared with daytime."

Page 16, line 14: "…low removal processes..."

Should this read: "…slow removal processes..." or "…limited removal processes..."?

[Figure]

Page 17, line 26: "Note that, South-Southwest wind directions were mainly encountered during nighttime"

Could it be that the profile observed for this factor is simply dictated by the wind direction rather than a change in local emissions? Perhaps an added comment here would help to clarify?

Page 18, line 13-29: This section is should be re-written for clarity.

I think the main point of this paragraph is that for two specific periods (when the air mass was originating from the southwest) factors 3, 4 and 5 (and other combustion tracers) were all well-correlated. The reason for this was the sizeable distance travelled from the source region and so would adversely affect the PMF analysis results. Hence, these periods were omitted from the subsequent analyses.

Page 18, line 33: "Factor 3 displays faire correlation with ethylene (r = 0.94). Factor 3 seems to correlate..."

Should read: "Factor 3 displays fair correlation with ethylene (r = 0.94) and seems to correlate..."

Page 19, line 2: "...increase in midmorning..."

Should this read: "...increase in mid-morning..."

Page 19, line 13-14: "...this source exhibits higher contributions during the day with peaks at 5 h - 6 h, 10 h, 12 h, 15 h LT corresponding to..."

The diel variation in figure 12 doesn't seem to agree with this statement. Can the authors please clarify what they are referring to here?

Page 19, line 17-23: When discussing the contributions to factors 4 and 5. Gasoline evaporation is included in both. More clarification is needed to explain why these factors are distinct from one another.

Page 20, line 25-26: "Hence, the high abundance of long-lived species in combination with the lack of shorter-lived compounds suggests here aged air masses transported towards the sampling site."

Should perhaps read: "Hence, the high abundance of long-lived species, in combination with the lack of shorter-lived compounds, suggests that aged air masses are being transported towards the sampling site."

Page 21, line 27-28: "...acetone (2.72 $\mu$g. m$-3$) with either biogenic origins (biogenic source 2) and primary/secondary anthropogenic origins..."

Should perhaps read: "...acetone (2.72 $\mu$g. m$-3$) with both biogenic origins (biogenic source 2) and primary/secondary anthropogenic origins..."

Page 22, line 5-6: "...to provide a comprehensive characterization of VOCs for the entire Mediterranean."

Not convinced that this can be described as comprehensive? I'd prefer this to be dropped to just read: "...to provide a characterization of VOCs for the entire Mediterranean."

Page 22, line 8: "On both sampling site, primary anthropogenic..."

Should read "At both sampling sites, primary anthropogenic..."

Page 22, line 9: "Similarly to our regional..."

Should read "Similar to our regional..."

Page 22, 4.1.2 Comparison with another PMF study performed at a remote site of the Mediterranean region:

Some of the differences described here appear to be (possibly) explained by differences in how the compounds were apportioned. Early in the manuscript the authors describe how (paraphrasing) the PMF analysis was tuned to give optimal results for this

site over this measurement period and presumably Michoud et al. have done similar for that study. Perhaps the only real way to make a valid comparison between the PMF results for the region would be to merge/re-assign the PMF factors to fit both sites? I can see that this may not be suitable for local pollution events, but would perhaps better describe the main factors affecting the "Mediterranean region". I don't expect the authors to actually do this here, but they could perhaps summarise the possible benefits and drawbacks of such a study.

Page 23, line 27: "...contributions were twice higher when..."

Should perhaps read "...contributions were twice as high when..."

Page 23, line 26-28: "The average OA contributions were twice higher when the station was under continental influence comparing to the one under marine influence."

I think this statement needs clarifying. From the look of the figure it depends which continent you refer to (Africa, Europe or Asia). Perhaps I am reading this incorrectly, but some clarity in the text would help here I think.

Tables, Figures and captions:

Table 1, caption: "Details of technics and measurements..."

Should read: "Details of techniques and measurements..."

Figure2: Can the resolution of the figure be improved? The axes and figure caption look slightly blurred to me.

Figure3: In the bottom right hand corner of the figure it states "calm = 0%"

The authors should clarify what this means and its significance, or remove it from the figure altogether.

Figure 5, caption: "Time series of a selection of anthropogenic VOCs (isoprene and $\alpha$-pinene – green lines)"

Should this be "biogenic"?

Figure7: Certain masses are reported and shown in the figure which presumably re-late to the PTRMS protonated masses. It would be useful to the reader to have the compound name(s) (accepting the uncertainty in exactly which compounds are being reported by the PTRMS), also listed in the figure.

Figure 8: Could a fifth panel be included in this figure for a stacked-area plot showing a time-series of the percentage contribution of each factor to the total? This would show very neatly which are the dominant sources factors at each point along the timeseries.

Figure 9, 12, 13, 18 and 19: A very minor point: Individual panes/plots are labelled as "a, b, c and d", but are described in the caption as "A, B, C, and D"

Please change these for consistency/clarity

Figure 10: Could this figure be moved into the Supplementary information section?

Figure 11: Perhaps it shows my lack of knowledge of the geography of the Mediter-anean region, but this figure is difficult for me to read. Is it possible to include some major country names as markers?

Could this figure be moved into the Supplementary information section? It is only rarely referred to in the text.

Supplementary Information

No 1,3 butadiene data is given in table SI-5 – was it measured? If so, it would be of interest.

---

## Referee Comment (RC2) · Anonymous Referee #2 · 23 Jun 2017

Comments on the Manuscript Titled "Origin and variability of volatile organic compounds observed at an Eastern Mediterranean background site (Cyprus)"

This manuscript presents the on-line measurements of 24 volatile organic compounds (VOCs) during a field campaign at a background site of Cyprus in March 2015. Based on the measurements, the temporal variability of VOCs was investigated. Six major sources and corresponding origins of VOCs were addressed by time series analysis, PMF receptor model, as well as CPF and CF. Furthermore, the influences of biogenic and anthropogenic sources on VOCs compositions were distinguished by a combined analysis of VOCs PMF factors with source apportioned OA. The work described in this

manuscript would definitely provide a better understanding of the air pollution of VOCs as well as their sources and fate impacting the Eastern Mediterranean region.

The comments on the manuscript are listed as follows.

1. Page 2-3: It would be better to shorten the "Introduction" part. For example, the second, third and fourth paragraphs in this part on page 2-3 would be shortened by removing certain general information.

2. Page 8 / Line 15-24: The "Off-line VOCs measurements" part on page 8 would be removed since data obtained by off-line measurement have not been used in this manuscript.

3. Page 8 / Line 27-30: The comparison of measurements between PTR-MS and GC-FID shows low intercept of 0.10 $\mu$g.m-3 for benzene and 0.13 $\mu$g.m-3 for toluene, as stated in the manuscript. However, the intercepts would not be low enough when the mean concentrations observed at the CAO in this campaign (0.37 $\mu$g.m-3 for benzene and 0.19 $\mu$g.m-3 for toluene) are considered.

4. Page 14 / Line 3: It is stated that the CAO was affected by air mass originating from "West of Turkey" (Page 14 / Line 3). But in the most part of the manuscript, it is stated that the CAO was affected by air mass originating from "South of Turkey". And in Figure 18, the factor contributions to VOCs were similar when air mass were originated from West of Turkey (C5) compared to from Marine (C2). Does this indicate that the "West of Turkey" is clean area?

5. Page 16 / Line 10-11: The interpretation of low concentration of $\alpha$-pinene in daytime would not be convincing. Both $\alpha$-pinene and isoprene can react with daytime oxidants with lifetime of 1.4h and 2.3h respectively (please see "3.5.1" part). But high concentration was observed for isoprene in daytime, while low concentration was observed for $\alpha$-pinene in daytime. Please provide more interpretation.

6. Page 20 / "3.5.3 Regional background (factor 6)": It would be suggested to add a

clear definition of "regional background". The definition would be helpful to understand the factor 6, since the source areas associated with factor 4, 5 and 6 are all South of Turkey or Southwest/Southeast of Turkey.

7. Page 23-25 / "4.2.2 Relationship between VOCs and OA": It would be suggested to add p value asssociated with correlation coefficient (r). With the p value, it would be more convincing to state that the correlation is statistically significant.

8. Page 46 / Figure 5: The word "anthropogenic" in the caption of Figure 5 would be "biogenic"; the word "m69" in the fourth drawing would be "isoprene".

---

## Author Comment (AC1) · 4 Aug 2017

Dear Referee #1, We would like to thank you for your general feedback and your useful comments/questions for improving the quality of this manuscript. All your comments were taken into account in the revised version of the manuscript. Authors' answers were uploaded as a *.pdf file and were displayed as a supplement to this comment.

Please also note the supplement to this comment:
https://www.atmos-chem-phys-discuss.net/acp-2016-1178/acp-2016-1178-AC1-supplement.pdf

[Figure]

[Figure]

**Supplement:**

**acp-2016-1178 : "Origin and variability of volatile organic compounds observed at an Eastern Mediterranean background site (Cyprus)"**

The article presents an extremely thorough and comprehensive overview of the field site on the Island of Cyprus. It details the analytical methods and instruments used; describes the chemical and broad aerosol composition; identifies the most likely sources of the pollutants observed; and describes the observations in the broader context of the "Mediterranean region". It provides a very good description of the site for anyone wishing to make observations in this region. The article, however, is (inevitably) very long and I fear that some of the impact of the paper may be lost due to its length. That said, this is no reason to exclude the paper from publication and so I recommend that it be published subject to the authors addressing the following comments.

**Authors' Responses to Referee #1**

We would like to thank the Referee #1 for her/his general feedback and each of her/his useful comments/questions for improving the quality of this manuscript. All comments addressed by both reviewers have been taken into account in the preparation of the revised version of the manuscript. In this respect, several figures were notably modified included in the supplementary. Please note that figures numbers are now different in this new version.

In the present document, authors' answers to the specific comments addressed by Referee #1 are mentioned in **blue**, while changes made to the revised manuscript are shown in *italic.*

**1/Specific comments:**

**1.1/** A wide range of VOCs are reported from the site and yet 1,3-butadiene a commonly reported anthropogenic compound which fits within the range of VOCs covered by the instruments used (and has important health and atmospheric chemistry impacts) does not appear to be reported. It would likely be of interest to many others in the field. Could this be included or were problems encountered for measurement of this compound?
1,3-butadiene was a targeted species in this study considering its impacts as already mentioned by reviewer #1. Calibrations were performed using a standard gas mixture containing 30 hydrocarbon species, including 1,3-butadiene, at a concentration level of 4 ppb certified by NPL (National Physical Laboratory, Teddington, Middlesex, UK). These species are recommended by the directive 2002/3/CE (12/02/02). 1,3-butadiene retention time and coefficient response were hence determined. However, 1,3-butadiene measured at CAO was most of the time below its detection limit and up to 0.09 µg.m$^{-3}$. This information is now included in the manuscript (Sect. I-5 in the supplement).

**1.2/** Some of the VOCs reported here are unsaturated compounds which are known to be highly reactive to ozone. There are known potential interferences caused when measuring these compounds in the atmosphere with GC systems. Were any measures taken to remove the ozone from the samples? If not, were the additional uncertainties

resulting from ozone reactions considered here? Perhaps inclusion of a short discussion of these issues could be added?

We didn't use any ozone scrubber for on-line measurements. However, as recommended by Detournay et al. (2011), different ozone scrubbers were used during the sampling of off-line measurements presented in section 2.2.1 in order to prevent any ozonolysis of the measured compounds. A KI ozone scrubber was installed upstream of the sampling onto DNPH cartridges, while a $MnO_2$ ozone scrubber was used for the multi-sorbent cartridges.

In addition to their on-line measurements, α-pinene and acetaldehyde were also measured by off-line techniques. α-Pinene was collected by multi-sorbent cartridges, analyzed after by GC-FID, while acetaldehyde was sampled on DNPH cartridges, analyzed after by HPLC-UV. α-Pinene and acetaldehyde were hence chosen to see the potential influence of ozone on on-line measurement by the cross-checking of the results during the field campaign. Correlation between on-line and off-line measurements of α-pinene and acetaldehyde concentrations displayed good determination coefficients (r: 0.83 and 0.90 for α-pinene and acetaldehyde, respectively). The slope is also close to one for both compounds (1.10 and 1.16 for α-pinene and acetaldehyde, respectively). As α-pinene and acetaldehyde on-line and off-line measurements have shown similar levels and variations, we think that potential interferences of ozone caused on VOCs measurements with GC systems were potentially limited in this study.

Correction applied in the revised manuscript:

Page 8, lines 15 – 26: "***VOCs intercomparison:***

*[…] The sum of pinenes measured by the GC-FID was compared to the (non speciated) monoterpenes measured by PTR-MS, yielding the same variability and consistent ranges of concentrations (r: 0.92 and slope: 0.96). The same conclusion was obtained for α-pinene, measured by GC-FID and off-line technique, and acetaldehyde, measured by PTR-MS and off-line technique. Correlation between on-line and off-line measurements of α-pinene and acetaldehyde concentrations displayed good determination coefficients (r: 0.83 and 0.90 for α-pinene and acetaldehyde, respectively). The slope is also close to one for both compounds (1.10 and 1.16 for α-pinene and acetaldehyde, respectively). Note that, no ozone scrubber was applied on GC systems to prevent any ozonolysis of the measured compounds. However, different ozone scrubbers were used during the sampling of off-line measurements as recommended by Detournay et al. (2011). The consistency of on-line measurements of α-pinene and acetaldehyde with off-line ones, in term of levels range and variation, ensured a limited interference of VOCs reaction with ozone on results derived from GC measurements.*

*As a result, recovery of the different techniques, regular quality checks and uncertainty determination approach have allowed to provide a good robustness of the dataset."*

**1.3/** The PMF analyses show a couple of gaps in the time-series around the 10[th] and 21[st] March (Figures 8, 14, 17 and 19). I may have missed it in the text, but the authors should confirm that this due to a loss of data (assuming it is) rather than being unable to assign these periods to any of the 6 factors identified.

Figure SI-1 in the supplement shows the period when each gas, aerosol and meteorological instrument was operating. As assumed by reviewer #1, gaps of 9-10 and 21 March were due to a loss of data of Chromatrap and of PTR-QMS, respectively. To run PMF, one needs to avoid any missing data. Rather than reconstructing VOC variations of these two days of missing data (which would have led to higher uncertainties on that days), the authors have preferred to select, as inputs of the PMF model, only the atmospheric data points when all the on-line VOCs instruments were available. The final chemical database consisted of 1,179 atmospheric data points which are sufficiently consequent to ensure robustness of the results.

**1.4/** Some of the differences described when comparing the PMF analyses from different site in the region appear to be (possibly) explained by differences in how the compounds were apportioned. Early in the manuscript the authors describe how (paraphrasing) the PMF analysis was tuned to give optimal results for this site over this measurement period and presumably Michoud et al. have done similar for that study. Perhaps the only real way to make a valid comparison between the PMF results for the region would be to merge/re-assign the PMF factors to fit both sites? I can see that this may not be suitable for local pollution events, but would perhaps better describe the main factors affecting the "Mediterranean region". I don't expect the authors to actually do this here, but they could perhaps summarize the possible benefits and drawbacks of such a study.

Few studies were dedicated to source apportionment of VOCs observed at remote/background sites and only two studies were performed at Mediterranean sites remote/background sites (Michoud et al. 2017 and this study). Additionally, only few of these studies included OVOC measurements. PMF studies performed at urban sites can be supported by near-field measurements having the fingerprint of specific source (e. g. Baudic et al. 2016). In remote/rural environments, VOCs result from direct emissions, chemistry, transport and mixing and therefore each individual factor cannot be attributed exclusively to one source category making potentially more difficult the interpretation of the result. Moreover, PMF analyses can be sensitive to the number of species included in the dataset, the instrumentation deployed, the time resolution of measurements, uncertainties of each measurement, the duration of the field campaign, period/season, but also the site typology. These parameters can potentially explain why some differences were observed in how the compounds were apportioned in the two PMF analyses, compared in Sect. 4.2.1 of the manuscript (Michoud et al. 2017 and this study).

Given the complexity of PMF analysis at remote/rural environments and in order to better characterize the sources of VOCs impacting each receptor site, two PMF analyses were willingly performed independently. We have chosen to not constrain the PMF factors to fit with Michoud et al. 2017 work to not take the risk to miss any important VOCs source specifically observed at CAO station.

Nevertheless, now that the VOCs sources impacting the CAO station were clearly characterized, the comparison with PMF analysis of Michoud et al 2017 show some common features, which could justify a general approach. Indeed, we have noticed that the classification of factors at both sites was linked to the difference in the sources

typology (biogenic vs anthropogenic) and/or the lifetime of compounds (short-lived, medium-lived and long-lived). Anthropogenic PMF factors were separated according to the lifetime of compounds which composed them, suggesting a homogeneity phenomenon on the entire basin, despite the different number of species of the two datasets.

In a future study, and following the idea of reviewer #1, it could be interesting to constrain some PMF factors (the ones relevant for larger scales than the local one) for a more comprehensive comparison of the VOCs sources impacting the Western and the Eastern Mediterranean. Nevertheless, we note that such a study would need an important step of data homogenization and preparation (selection of the common species, selection of factors to be constrained and associated reference profile...). As this idea goes beyond the scope of this paper, we have not discussed this point in the manuscript.

**2/Technical comments/corrections:**

**2.1/ Page 2, line 29:** "A robust tool to identify emission sources is Positive Matrix Factorization (PMF - Paatero, 1997; Paatero and Tapper, 1994) is one of the various tools developed to identify emission sources. Over the last decade ..."
Should read something like: "A robust tool to identify emission sources is Positive Matrix Factorization (PMF - Paatero, 1997; Paatero and Tapper, 1994). Over the last Decade ..."
Correction applied in the revised manuscript (Page 2, lines 19 – 20): *"A robust tool to identify emission sources is Positive Matrix Factorization (PMF - Paatero, 1997; Paatero and Tapper, 1994)."*

**2.2/ Page 3, line 22:** "... and especially its evolution ..." Should read: "... and especially their evolution ..."
Correction applied in the revised manuscript (Page 3, lines 6 – 7): *"It is, therefore, essential to understand the sources and fate of VOCs in the atmosphere, and especially their evolution during long-range transport."*

**2.3/ Page 3, line 24:** "Affected by important pollution sources, the Mediterranean is a sensitive region affected by both particulate and gaseous pollutants." Should read something like: "The Mediterranean is a sensitive region affected by both particulate and gaseous pollutants."
Correction applied in the revised manuscript (Page 3, line 8): *"The Mediterranean is a sensitive region affected by both particulate and gaseous pollutants."*

**2.4/ Page 4, line 17:** "... (geographical origin, fast/low transport) ..." Should read: "... (geographical origin, fast/slow transport) ..."

**Correction applied in the revised manuscript (Page 3, lines 32 - 34):** *"The period under investigation offered contrasted conditions in terms of air mass transport (geographical origin, fast/slow transport) and weather (temperature, humidity, precipitations...)."*

**2.5/ Page 4, line 19:** "It is ideally located, close from the coast" Should read: "It is ideally located, close to the coast"
**Correction applied in the revised manuscript (Page 4, lines 1 - 2):** *"It is ideally located, close to the coast and far from major Cypriot anthropogenic areas."*

**2.6/ Page 4, line 23:** "Our study performed in the Eastern Mediterranean will therefore offer a unique opportunity to provide a comprehensive characterization of VOCs for the entire Mediterranean." This is a very grand statement and, I would suggest, is overstating the robustness of the study somewhat. Perhaps "detailed" rather than "comprehensive" would be more accurate?
**Correction applied in the revised manuscript (Page 4, lines 5 - 6):** *"Our study will therefore offer a unique opportunity to characterize VOCs in the Eastern Mediterranean."*

**2.7/ Page 4, line 30:** "… performed at an another background site of …" Should read: "… performed at another background site of …"
**Correction applied in the revised manuscript (Page 4, lines 10 - 11):** *"Then, in Sect. 4.1, we compare VOCs concentrations measured during this intensive field campaign with previous measurements performed at another background site of Cyprus …"*

**2.8/ Page 5, line 12:** "… European Research Infrastructure fir the observation of …" Should read: "… European Research Infrastructure for the observation of …"
**The typo was corrected in the revised manuscript (Page 4, line 26).**

**2.9/ Page 5, line 17:** "… more than 35 km of main Cypriot agglomerations, with very poor influences of these anthropogenic emission areas." Do the authors mean "limited influences"?
**The authors mean "limited influences" to insist on the typology of the sampling site.**
**Correction applied in the revised manuscript (Page 4, lines 29 - 31):** *"The station is located in the central area of the island about 20 km from the coast and more than 35 km of main Cypriot agglomerations, with limited influences of these anthropogenic emission areas."*

**2.10/ Page 6, line 2:** "AirmoVOC C6-C12, the measurement of C6-C10 hydrocarbons"

Why were measurements not made up to C12? Perhaps a statement could be made in the text for clarity here?

The instrument is called AirmoVOC C6-C12 since measurement of VOCs from hexane to dodecane was ensured by the manufacturer Chromatotec. However, heavier compounds of the NPL standard were $C_{10}$ hydrocarbons and hence AirmoVOC C6-C12 measurements were not made up to $C_{12}$ hydrocarbons. Note that, dodecane was measured by off-line technique and its concentrations were of 0.02 µg.m-3 on average. For clarity, the name C6C12 has now been removed in the revised manuscript.

Correction applied in the revised manuscript (Page 5, lines 13 - 14): *"The first analyzer, ChromaTrap, was used for the measurement of $C_2$-$C_6$ hydrocarbons (alkanes, alkenes and alkynes) and the second, AirmoVOC, for the measurement of $C_6$-$C_{10}$ hydrocarbons (alkanes, monoterpenes and aromatic compounds)."*

**2.11/ Page 13, line 21:** "The most favorable conditions for high levels of VOCs (high temperatures, clear skies and low winds) were ..."

I disagree with this statement as it is written. Warmer temperatures may lead to greater emissions from biogenic and also evaporative sources, however emissions from heating sources etc are normally observed to decrease. Warmer conditions are also commonly associated with greater boundary layer height leading to lower concentrations.
Can the authors re-phrase this for improved clarity?

The authors decided only to discuss here of the best conditions to observe high biogenic VOCs emissions since, at this point of the paper, anthropogenic sources observed at CAO are not yet identified.

Correction applied in the revised manuscript (Page 13, lines 9 - 10): *"The most favorable conditions for high biogenic VOCs emissions (high temperatures, clear skies and low winds) were observed from 8 to 10 March and from 25 to 27 March."*

**2.12/ Page 13, line 23:** "... and rainy periods may participate to a larger development of vegetation." I suggest changing "participate" to "contribute"

Correction applied in the revised manuscript (Page 13, lines 10 - 12): *"Some rough-weather days, characterized by lower temperatures, heavy rain and strong winds also occurred (11 – 14 March, 20 – 22 March and 28 March) and rainy periods may contribute to a larger development of vegetation."*

**2.13/ Page 16, line 10:** "... elevated during nighttime than during the daytime." Should read: "... elevated during nighttime compared with daytime."

Correction applied in the revised manuscript (Page 15, lines 27 - 28): *"Surprisingly, α-pinene concentrations were elevated during nighttime compared with daytime."*

**2.14/ Page 16, line 14:** "... low removal processes ..." Should this read: "... slow removal processes ..." or "... limited removal processes ..."?

Correction applied in the revised manuscript (Page 15, lines 30 - 31): *"These nocturnal maxima were enhanced by the shallow nocturnal boundary layer, and the slow removal processes (i. e. low concentrations of oxidizing species) leading to higher concentrations."*

**2.15/ Page 17, line 26:** "Note that, South-Southwest wind directions were mainly encountered during nighttime"
Could it be that the profile observed for this factor is simply dictated by the wind direction rather than a change in local emissions? Perhaps an added comment here would help to clarify?

We agree with the reviewer that prevailing winds from South-Southwest (region of pine and oak forest) during the night can partly explain the high values of monoterpenes measured during the night. Nevertheless, this was not the only factor and we have noticed that monoterpenes variation was also dependent on vegetation type, and removal processes:

- The monoterpenes average diurnal patterns indicated that their emissions were solely dependent on temperature (Laothawornkitkul et al., 2009) and lower, but still significant emissions occurred throughout the night. This pattern was attributed to nocturnal emissions from monoterpenes storing plants from the understorey vegetation (Niinemet et al., 2004; Laothawornkitkul et al., 2009; Schurgers et al., 2009).
- These nocturnal maxima were also enhanced by the low removal processes (i.e. low concentrations of oxidizing species) and the shallow nocturnal boundary layer.

Correction applied in the revised manuscript (Page 17, lines 9 - 18): *"The diurnal profile of factor 1 exhibits higher contributions during nighttime, 18 h – 5 h LT (local time) (in agreement with the diurnal variability of α-pinene investigated in Sect. 3.4.2) and CPF analysis localizes this source in the South and Southwest directions from the sampling site, that were mainly encountered during nighttime (Sect. SI-3 in the supplement)". In these directions, the area corresponds to pines and/or oaks forests (Sect. SI-6 in the supplement - Fall, 2012), known as high emitters of pinenes but also OVOCs (e.g. acetone, Janson and de Serves, 2001). As a result, air masses observed at CAO during nighttime were enriched during transport over oak and pine forests with biogenic nocturnal emissions from plants with BVOCs storage compartments like coniferous species (Laothawornkitkul et al., 2009; Niinemet et al., 2004). These nocturnal maxima were also enhanced by the low removal processes (i.e. low concentrations of oxidizing species) and the shallow nocturnal boundary layer. "*

References:

Fall, P. L.: Modern vegetation, pollen and climate relationships on the Mediterranean island of Cyprus, Rev. Palaeobot. Palynol., 185, 79–92, doi:10.1016/j.revpalbo. 2012.08.002, 2012.

Janson, R. and de Serves, C.: Acetone and monoterpene emissions from the boreal forest in northern Europe, Atmos. Environ., 35(27), 4629–4637, doi:10.1016/S1352-2310(01)00160-1, 2001.

Laothawornkitkul, J., Taylor, J. E., Paul, N. D. and Hewitt, C. N.: Biogenic volatile organic compounds in the Earth system, New Phytol., 183(1), 27–51, doi:10.1111/j.1469-8137.2009.02859.x, 2009.

Niinemets, Ü., Loreto, F. and Reichstein, M.: Physiological and physicochemical controls on foliar volatile organic compound emissions, Trends Plant Sci., 9(4), 180–186, doi:10.1016/j.tplants.2004.02. 006, 2004.

Schurgers, G., Arneth, A., Holzinger, R. and Goldstein, A. H.: Process-based modelling of biogenic monoterpene emissions combining production and release from storage, Atmos Chem Phys, 9(10), 3409–3423, doi:10.5194/acp-9-3409-2009, 2009.

**2.16/ Page 18, lines 13 - 29:** This section is should be re-written for clarity.

I think the main point of this paragraph is that for two specific periods (when the air mass was originating from the southwest) factors 3, 4 and 5 (and other combustion tracers) were all well-correlated. The reason for this was the sizeable distance travelled from the source region and so would adversely affect the PMF analysis results. Hence, these periods were omitted from the subsequent analyses.

Following reviewer's suggestions, we have made efforts to re-write a revised version of the paragraph for clarity. In the revised manuscript, it now reads (Page 18, lines 6 - 21): *"Firstly, we note that these three anthropogenic factors showed enhanced contributions and similar variations during the periods when the station was influenced by air-masses imported from Southwest Asia (i. e. 6 to 12 March and from 26 to 29 March). These periods were also associated with enhanced levels of the anthropogenic compounds investigated in Sect. 3.4.1 but also $C_6$-$C_{14}$ alkanes and fossil fuel combustion tracers (CO, $NO_2$ and BC). These indications suggest that anthropogenic VOCs were potentially of the same origin when the station was influenced by air masses from Southwest Asia and this independently of their specific sources. To determine the potential origin of these anthropogenic events observed at CAO, the CF results, concerning the factor contributions associated to Southwest Asian air masses alone (SI-7 in the supplement), were investigated. These results pinpoint the Southeast coasts of Turkey as potential origin of these anthropogenic events observed at CAO. This region corresponds to densely populated areas of Turkey (including Adana and Gaziantep, with more than 1.6 million of inhabitants, the fifth and the sixth most densely populated cities in Turkey, respectively) with expected high anthropogenic emissions due to intense industrial and maritime activities (e.g. the seaport of Mersin) and a dense road network. As a conclusion, Southwest Asian air masses were associated with higher contributions of the three primary anthropogenic VOCs factors. Therefore, to study more specifically each of these three anthropogenic factors and their additional local/regional origins, they will be investigated in the rest of this section omitting the contributions associated to Southwest Asian air masses. Diel variabilities are shown in Fig. 10. CPF results were also investigated and are presented in Fig. 11."*

Furthermore, a higher continental influence on anthropogenic factors was noticed when the site received air masses coming from the Eastern Mediterranean region (i. e. NW Asia and SW Asia) than the Western region. Hence, this finding could be linked with the difference of distance between the sampling site and respective potential

emission areas of the Eastern/Western Mediterranean (as depicted by CF results plotted in Fig. 14 of the revised manuscript – note that, potential emissions areas were the same for factors 3 to 6). Indeed, Crete and Peloponnese regions, potential emissions areas attributed to the Western Mediterranean region influence of the site, are far from the sampling station of 600 km – 950 km while South and Southwest Turkish coasts, potential emissions areas attributed to the Eastern Mediterranean Region, are much closer from the site (100 km - 250 km from CAO). As a consequence, the shorter distance of potential Eastern Mediterranean emissions areas could lead to an injection of fresher anthropogenic sources in air masses observed at the receptor site. This suggestion is also consistent with the fact that even VOCs of short/medium lifetimes were influenced by this event despite the transport time to reach the receptor site. As a result, the authors think this regional influence is due to the relatively short distance travelled from the source region.

Correction applied in the revised manuscript (Page 23, lines 21 - 27): *"A higher continental influence on anthropogenic factors was noticed when the site received air masses coming from the Eastern Mediterranean region (i. e. NW Asia and SW Asia) than the Western region (i. e. Europe) as expected from the distance which separates the source region and the measuring site. Indeed, potential emissions areas attributed to the Western Mediterranean region influence of the site (as depicted by CF results plotted in Fig. 14), are far from the sampling station of 600 km – 950 km while South Turkish coasts are much closer from the site (100 km - 250 km). As a consequence, the shorter distance of potential Eastern Mediterranean emissions areas could lead to an injection of fresher anthropogenic sources in air masses observed at CAO."*

[Figure]

**Figure 14 of the revised manuscript: Potential source areas contributing to the VOCs factor 6 in function of air mass origins. Contributions are in units of µg.m$^{-3}$. All – without distinction of air mass origins; C2 – marine air masses; C3 – Europe; C4 – NW Asia; C5 – West of Turkey; C7 – SW Asia. Low numbers of samples associated to clusters 0 and 1 (Local and N. Africa, respectively – Figure 2) don't allow to apply CPF analysis only considering these air masses origin.**

**2.17/ Page 18, line 33:** "Factor 3 displays faire correlation with ethylene (r = 0.94). Factor 3 seems to correlate …" Should read: "Factor 3 displays fair correlation with ethylene (r = 0.94) and seems to correlate …"

Correction applied in the revised manuscript (Page 18, lines 25 - 26): "*Factor 3 displays faire correlation with ethylene (r = 0.94) and seems to correlate with NO$_2$, CO and BC, which are known to be relevant vehicle exhaust markers (r = 0.41, 0.40, 0.37, respectively).*"

**2.18/ Page 19, line 2:** "… increase in midmorning …" Should this read: "… increase in mid-morning …"

Correction applied in the revised manuscript (Page 18, lines 26 - 27): *"Even if the diurnal profile does not exhibit a clear variability except a slight increase in mid-morning (Fig. 10), the time series shows a scattered variability (Fig. 8)."*

**2.19/ Page 19, lines 13 - 14:** "… this source exhibits higher contributions during the day with peaks at 5 h - 6 h, 10 h, 12 h, 15 h LT corresponding to …" The diel variation in figure 12 doesn't seem to agree with this statement. Can the authors please clarify what they are referring to here?

During the field campaign, we have noticed circulation of vehicles on the hill which could explain the higher contributions of factor 4 at some times. Nevertheless, for clarity, we have modified the manuscript as followed (Page 19, lines 5 - 7):

*"The diurnal variation of this source exhibits higher contributions during the day with peaks during the period 10h- 15 h LT corresponding to typical circulation of vehicles on the hill where the station is located."*

**2.20/ Page 19, lines 17 - 23:** When discussing the contributions to factors 4 and 5. Gasoline evaporation is included in both. More clarification is needed to explain why these factors are distinct from one another.

One hand, as PMF analysis was conducted on VOCs dataset collected at a background site, some of the computed factors may not be directly related to emission profiles but should rather be interpreted as aged profiles originating from different sources belonging to similar source categories (Sauvage et al., 2009). Hence, we have identified factors 4 and 5 as evaporative sources instead of only gasoline evaporation.

On the other hand, despite their similar variations, the PMF model assigned i, n-butanes and i, n-pentanes into two different factors for any PMF solution of at least 4 factors. Considering the optimized PMF solution presented in the paper, factor 4 was mainly composed of i,n-butanes and factor 5 of i,n-pentanes and toluene. Butanes concentrations have shown higher background levels than pentanes and toluene consistent with their respective lifetime (5-6 days and 2-3 days for i,n-butanes and i, n-pentanes/toluene, respectively) that may partly explain the distinction of these two factors. Additionally, higher contributions were noticed during daytime for factor 4 (CPF plots depicted in Fig. 11 of the revised manuscript) and originating from North, Northeast and East wind sectors. Factor 5 has shown a diurnal variability (7 h – 18 h LT) in agreement with factor 4 (r= 0.64) when the diel profile did not include contributions obtained when winds were of South and Southeast directions (diurnal profile of factor 5 presented in Sect. SI-9). What distinguish factor 5 from factor 4 could also be the fact that factor 5 was influenced by an additional geographical location compared to factor 4. Indeed, the diurnal pattern of factor 5 (Fig. 10 of the revised manuscript) has also shown high contributions during nighttime when the station was under the influence of wind coming from the South and the Southeast directions and that could be associated to industrial sources. The paragraph presenting factor 5 was modified in the revised manuscript to better highlight what distinguish factor 5 from factor 4 (Page 19, lines 12 - 20).

**2.21/ Page 20, lines 25 - 26:** "Hence, the high abundance of long-lived species in combination with the lack of shorter-lived compounds suggests here aged air masses transported towards the sampling site." Should perhaps read: "Hence, the high abundance of long-lived species, in combination with the lack of shorter-lived compounds, suggests that aged air masses are being transported towards the sampling site."

Correction applied in the revised manuscript (Page 20, lines 20 - 21): *"Hence, the high abundance of long-lived species, in combination with the lack of shorter-lived compounds, suggests here aged air masses transported towards the sampling site."*

**2.22/ Page 21, lines 27 - 28:** "… acetone (2.72 µg. m$^{-3}$) with either biogenic origins (biogenic source 2) and primary/secondary anthropogenic origins …" Should perhaps read: "… acetone (2.72 µg. m$^{-3}$) with both biogenic origins (biogenic source 2) and primary/secondary anthropogenic origins …"

Correction applied in the revised manuscript (Page 21, lines 24 - 27): *"In this study, we found more pronounced OVOCs diel variations (not shown in this article but similar to biogenic source 2 diel variation), comparable average concentration of methanol (3.84 µg.m$^{-3}$) with mostly biogenic origins and lower concentration of acetone (2.72 µg.m$^{-3}$) with both biogenic origins (biogenic source 2) and primary/secondary anthropogenic origins (regional background)."*

**2.23/ Page 22, lines 5 - 6:** "… to provide a comprehensive characterization of VOCs for the entire Mediterranean." Not convinced that this can be described as comprehensive? I'd prefer this to be dropped to just read: "… to provide a characterization of VOCs for the entire Mediterranean."

Correction applied in the revised manuscript (Page 22, lines 3 - 5): *"They have performed a PMF analysis on a gas database made of 42 VOCs, including primary VOCs with anthropogenic and biogenic origins and OVOCs and therefore offer a unique opportunity to provide a characterization of VOCs for the entire Mediterranean."*

**2.24/ Page 22, line 8:** "On both sampling site, primary anthropogenic …" Should read "At both sampling sites, primary anthropogenic …"

Correction applied in the revised manuscript (Page 22, lines 8 - 9): *"At both sampling site, primary anthropogenic PMF factors were separated according to the lifetime of compounds which composed them suggesting a homogeneity phenomenon on the entire basin."*

**2.25/ Page 22, line 9:** "Similarly to our regional …" Should read "Similar to our regional …"

Correction applied in the revised manuscript (Page 22, lines 9 - 10): *"Similar to our regional background factor, their "long-lived anthropogenic" factor was mainly composed of ..."*

**2.26/ Page 22, line 9:** Comparison with another PMF study performed at a remote site of the Mediterranean region:
Some of the differences described here appear to be (possibly) explained by differences in how the compounds were apportioned. Early in the manuscript the authors describe how (paraphrasing) the PMF analysis was tuned to give optimal results for this site over this measurement period and presumably Michoud et al. have done similar for that study. Perhaps the only real way to make a valid comparison between the PMF results for the region would be to merge/re-assign the PMF factors to fit both sites? I can see that this may not be suitable for local pollution events, but would perhaps better describe the main factors affecting the "Mediterranean region". I don't expect the authors to actually do this here, but they could perhaps summarize the possible benefits and drawbacks of such a study.
Please refer to the previous comment 1.4.

**2.27/ Page 23, line 27:** "... contributions were twice higher when" Should perhaps read "... contributions were twice as high when ..."
Correction applied in the revised manuscript (Page 24, lines 5 - 6): *"The average OA contributions were twice as high when the station was under continental influence ..."*

**2.28/ Page 23, lines 26 - 28:** "The average OA contributions were twice higher when the station was under continental influence comparing to the one under marine influence."
I think this statement needs clarifying. From the look of the figure it depends on which continent you refer to (Africa, Europe or Asia). Perhaps I am reading this incorrectly, but some clarity in the text would help here I think.
The authors refer here to Asia and Europe continental contribution. Note that, air masses originating from Africa were little observed at CAO during March 2015 (2 % - Fig. 2 of the revised manuscript).
Correction applied in the revised manuscript (Page 24, lines 5 - 7): *"The average OA contributions were twice as high when the station was under continental influence (i. e. Europe and Asia) comparing to the one under marine influence."*

**3/Tables, Figures and captions:**
**3.1/ Table 1, caption:** "Details of technics and measurements…" Should read: "Details of techniques and measurements…".
Correction applied in the revised manuscript: *"Details of techniques and measurements".*

**3.2/ Figure 2:** Can the resolution of the figure be improved? The axes and figure caption look slightly blurred to me.

Figure 2 was modified in the revised manuscript:

[Figure]

**Figure 2 (of the revised manuscript): Classification of air masses which impacted the site during the intensive field campaign of March 2015 and their relative contribution. A fraction of 2 % (not shown here) is attributed to air masses of mixed origins.**

**3.3/ Figure 3:** In the bottom right hand corner of the figure it states "calm = 0%". The authors should clarify what this means and its significance, or remove it from the figure altogether.

The authors removed it from figure 3 but also from the figure of Sect. SI-3 in the supplement.

**3.4/ Figure 5, caption:** "Time series of a selection of anthropogenic VOCs (isoprene and pinene – green lines)". Should this be "biogenic"?

Correction applied in the revised manuscript: *"Time series of a selection of biogenic VOCs (isoprene and pinene – green lines)".*

**3.5/ Figure 7:** Certain masses are reported and shown in the figures which presumably relate to the PTRMS protonated masses. It would be useful to the reader to have the compound name(s) (accepting the uncertainty in exactly which compounds are being reported by the PTRMS), also listed in the figure.

Figure 7 was modified in the revised manuscript:

[Figure]

**Figure 7 (of the revised manuscript): Chemical profiles of the 6-factor PMF solution (20 VOCs). The contribution of the factor to each specie ($\mu g. m^{-3}$) and the percent of each specie apportioned to the factor are displayed as a grey bar and a color circle, respectively. Factor 1 - biogenic source 1; factor 2 - biogenic source 2; factor 3 – short-lived combustion source; factor 4 – evaporative sources; factor 5 – industrial and evaporative sources; factor 6 – regional background.**

**3.6/ Figure 8:** Could a fifth panel be included in this figure for a stacked-area plot showing a time-series of the percentage contribution of each factor to the total? This would show very neatly which are the dominant sources factors at each point along the time series.

Figure 8 and its caption were modified in the revised manuscript:

[Figure]

**Figure 8 (of the revised manuscript): Time series of VOCs factor contributions (a - $\mu g.\,m^{-3}$) and accumulated relative VOCs contributions (b). Factor 1 - biogenic source 1; factor 2 - biogenic source 2; factor 3 – short-lived combustion source; factor 4 – evaporative sources; factor 5 – industrial and evaporative sources; factor 6 – regional background.**

**3.7/ Figure 9:** A very minor point: Individual panes/plots are labelled as "a, b, c and d", but are described in the caption as "A, B, C, and D". Please change these for consistency/clarity.
Individual panes/plots are described in the caption as "a, b, c and d" in the revised manuscript.

**3.8/ Figure 10:** Could this figure be moved into the Supplementary information section?
Figure 10 was moved into the Supplementary information section (SI-6) in the revised manuscript.

**3.9/ Figure 11:** Perhaps it shows my lack of knowledge of the geography of the Mediterranean region, but this figure is difficult for me to read. Is it possible to include some major country names as markers? Could this figure be moved into the Supplementary information section? It is only rarely referred to in the text.

Figure 11 was moved into the Supplementary information section (Sect. SI-7) in the revised manuscript. Major country and cities names were now indicated on the figure of section SI-7 of the supplement materials:

**SI-7 Potential source areas contributing to the 3 anthropogenic VOCs factors, determined using the CF model 5-days back-trajectories from HYSPLIT model, as a function of air masses origin.**

Contributions are in µg.m$^{-3}$. Cluster 0 – Local; Cluster 1 – N. Africa; Cluster 2 – marine air masses; Cluster 3 – Europe; Cluster 4 – NW Asia; Cluster 5 – West of Turkey; Cluster 7 – SW Asia.

[Figure]

**4/Supplementary information**

**4.1/**No 1,3 butadiene data is given in table SI-5 – was it measured? If so, it would be of interest.

Please refer to the previous comment 1.1.

---

## Author Comment (AC2) · 4 Aug 2017

Dear Referee #2, We would like to thank you for your general feedback and your useful comments/questions for improving the quality of this manuscript. All your comments were taken into account in the revised version of the manuscript. Authors' answers were uploaded as a *.pdf file and were displayed as a supplement to this comment.

Please also note the supplement to this comment:
https://www.atmos-chem-phys-discuss.net/acp-2016-1178/acp-2016-1178-AC2-supplement.pdf

[Figure]

[Figure]

**Supplement:**

**acp-2016-1178 : "Origin and variability of volatile organic compounds observed at an Eastern Mediterranean background site (Cyprus)"**

This manuscript presents the on-line measurements of 24 volatile organic compounds (VOCs) during a field campaign at a background site of Cyprus in March 2015. Based on the measurements, the temporal variability of VOCs was investigated. Six major sources and corresponding origins of VOCs were addressed by time series analysis, PMF receptor model, as well as CPF and CF. Furthermore, the influences of biogenic and anthropogenic sources on VOCs compositions were distinguished by a combined analysis of VOCs PMF factors with source apportioned OA. The work described in this manuscript would definitely provide a better understanding of the air pollution of VOCs as well as their sources and fate impacting the Eastern Mediterranean region.

**Authors' Responses to Referee #2**

We would like to thank the Referee #2 for her/his general feedback and each of her/his useful comments/questions for improving the quality of this manuscript. All comments addressed by both reviewers have been taken into account in the preparation of the revised version of the manuscript. In this respect, several figures were notably modified and in the supplementary. Please note that figures numbers are now different in this new version.

In the present document, authors' answers to the specific comments addressed by Referee #2 are mentioned in **blue**, while changes made to the revised manuscript are shown in *italic.*

The comments on the manuscript are listed as follows:

**1/ Pages 2 - 3:** It would be better to shorten the "Introduction" part. For example, the second, third and fourth paragraphs in this part on page 2-3 would be shortened by removing certain general information.

Following reviewer's suggestions, we have made efforts to write a revised version of the introduction with conciseness. In the revised manuscript, it now reads (Pages 2 - 3):
"**

[revised manuscript text omitted]

**2/ Page 8, lines 15 - 24:** The "Off-line VOCs measurements" part on page 8 would be removed since data obtained by off-line measurement have not been used in this manuscript.

Off-line VOCs measurements were used in order to consolidate the robustness of the PMF dataset.

In the revised manuscript, and following a comment from reviewer #1, a short section has been added to give the results about the comparison between on- and off-line measurements.

Corrections applied in the revised manuscript:

Page 8, lines 1 - 4: *"**Off-line VOCs measurements:**

[…] Here, VOCs measured off-line were used as independent parameters to consolidate the robustness of on-line measurements by inter-comparison of VOCs measured by different techniques. They will be further presented and investigated in another paper dedicated on biogenic and oxygenated VOCs."*

Page 8, lines 15 - 26: *"**VOCs intercomparison:**

[…] The sum of pinenes measured by the GC-FID was compared to the (non speciated) monoterpenes measured by PTR-MS, yielding the same variability and consistent ranges of concentrations (r: 0.92 and slope: 0.96). The same conclusion was obtained for α-pinene, measured by GC-FID and off-line technique, and acetaldehyde, measured by PTR-MS and off-line technique. Correlation between on-line and off-line measurements of α-pinene and acetaldehyde concentrations displayed good determination coefficients (r: 0.83 and 0.90 for α-pinene and acetaldehyde, respectively). The slope is also close to one for both compounds (1.10 and 1.16 for α-pinene and acetaldehyde, respectively). Note that, no ozone scrubber was applied on GC systems to prevent any ozonolysis of the measured compounds. However, different ozone scrubbers were used during the sampling of off-line measurements as recommended by Detournay et al. (2011). The consistency of on-line measurements of α-pinene and acetaldehyde with off-line ones, in term of levels range and variation, ensured a limited interference of VOCs reaction with ozone on results derived from GC measurements.*

*As a result, recovery of the different techniques, regular quality checks and uncertainty determination approach have allowed to provide a good robustness of the dataset."*

**3/ Page 8, lines 27 - 30:** The comparison of measurements between PTR-MS and GC-FID shows low intercept of 0.10 µg.m-3 for benzene and 0.13 µg.m-3 for toluene, as stated in the manuscript. However, the intercepts would not be low enough when the mean concentrations observed at the CAO in this campaign (0.37 µg.m-3 for benzene and 0.19 µg.m-3 for toluene) are considered.

We agree with the reviewer that the intercepts are relatively high in comparison with the mean concentrations. Note that, the two instruments had different time resolution: for PTR-MS, signals of every masses were acquired every 10 min with a dwell time of 5 s

per mass and, for AirmoVOC, each measurement of 30 min started with an analysis period of 22.5 min. Additionally, the lowest concentrations were generally entailed by higher relative uncertainties than the mean/high concentrations.

In order to nuance that point, we have decided to replace "low" by "acceptable" in the revised manuscript (Page 8, line 10).

**4/ Page 14, line 3:** It is stated that the CAO was affected by air mass originating from "West of Turkey" (Page 14, line 3). But in the most part of the manuscript, it is stated that the CAO was affected by air mass originating from "South of Turkey". And in Figure 18, the factor contributions to VOCs were similar when air masses were originated from West of Turkey (C5) compared to from Marine (C2). Does this indicate that the "West of Turkey" is clean area?

On one hand, in the manuscript, "West of Turkey" refers to a source region (i. e. cluster 5) of the classification of air-mass origins (Fig. 2 of the revised manuscript) based on the analysis of the retroplumes. "South of Turkey" refers to potential origin of regional contribution observed at CAO when air masses were originated from both the Southwest Asia and the Northwest Asia (i. e. clusters 7 and 4, respectively). Potential emissions areas were pinpointed by concentration field (CF) results of anthropogenic factors. CF method consists in redistributing concentrations of a variable observed at a receptor site along the back-trajectories, ending at this site, inside a gridded map. The CAO station was mostly influenced by continental air masses originating from "Southwest Asia" (cluster 7 – 31 %) and "Northwest Asia" (cluster 4 – 28 %) while it was only influenced 10 % of the field campaign by "West of Turkey" air masses (cluster 5). These findings explain why it is stated that the CAO was affected by air mass originating from West of Asia. These air masses have transported toward the site regional contribution of potential emission from the South coasts of Turkey area.

On the other hand, a higher continental influence was noticed on anthropogenic factors when the site received air masses coming from the West of Asia than the West of Turkey (Fig. 16 of the revised manuscript). Potential emissions areas associated to air masses originating from the West of Turkey were the Southwest coasts of Turkey (as depicted by CF results of factor 6 plotted in Fig. 14 of the revised manuscript– note that, potential emissions areas were the same for factors 3 to 6). Additional more distant emissions areas of the West of Turkey were not observed by CF plots, such as the Istanbul region, which corresponds to the most densely populated areas of Turkey, with expected high anthropogenic emissions. Istanbul is far from CAO station of 700 km while the South coasts of Turkey are much closer from the site (100 km - 250 km from CAO). As a consequence, the shorter distance of potential emissions areas, associated to the South coasts of Turkey, could lead to an injection of fresher anthropogenic sources in air masses observed at the receptor site. Furthermore, the Southwest coasts of Turkey correspond to less densely populated areas compared to the Southeast coasts of Turkey (potential emissions area associated to air masses originating from the West of Asia – Fig. 14 of the revised manuscript). These findings only indicate that CAO is not

influenced by important anthropogenic emissions areas when air masses were originated from the West of Turkey and cannot suggest that the "West of Turkey" is a clean area.

Corrections applied in the revised manuscript (Page 23, lines 28 - 33):

*"Additionally, a higher continental influence was noticed on anthropogenic factors when the site received air masses coming from the West of Asia than the West of Turkey. Potential emissions areas associated to air masses originating from the West of Turkey were the Southwest coasts of Turkey (as depicted by CF results plotted in Fig. 14), corresponding to less densely populated areas compared to the Southeast coasts of Turkey. Note that, additional more distant emissions areas of the West of Turkey, such as the Istanbul region with expected high anthropogenic emissions, were not observed by CF plots that could explain why CAO was not influenced by important anthropogenic emissions areas when air masses were originated from the West of Turkey."*

[Figure]

**Figure 14 (of the revised manuscript): Potential source areas contributing to the VOCs factor 6 in function of air mass origins. Contributions are in units of µg.m⁻³. All – without distinction of air mass origins; C2 – marine air masses; C3 – Europe; C4 – NW Asia; C5 – West of Turkey; C7 – SW Asia. Low numbers of samples associated to clusters 0 and 1 (Local and N. Africa, respectively – figure 2) don't allow to apply CPF analysis only considering these air masses origin.**

[Figure]

**Figure 16 (of the revised manuscript):** Accumulated average contributions of the OA and VOCs factors (figures a and b, respectively) in function of air mass origins. Classification of air masses: C0 - Local; C1 - N. Africa; C2 - Marine; C3 - Europe; C4 - NW. Asia; C5 - W. of Turkey and C7 - SW. Asia. VOCs factors: factor 1 - biogenic source 1; factor 2 - biogenic source 2; factor 3 – short-lived combustion source; factor 4 – evaporative sources; factor 5 – industrial and evaporative sources; factor 6 – regional background. OA factors: HOA - hydrogen-like OA; SV-OOA – semi-volatile oxygen-like OA; LV-OOA – low-volatile oxygen-like OA.

**5/ Page 16, lines 10 - 11:** The interpretation of low concentration of α-pinene in daytime would not be convincing. Both α-pinene and isoprene can react with daytime oxidants with lifetime of 1.4h and 2.3h respectively (please see "3.5.1" part). But high concentration was observed for isoprene in daytime, while low concentration was observed for α-pinene in daytime. Please provide more interpretation

We have decided to remove the phrase "A possible interpretation could be that α-pinene was rapidly consumed by daytime oxidants due to its high reactivity." In the revised manuscript, the interpretation of monoterpenes variability is explained by nocturnal monoterpenes emissions, as other studies, since some additional elements were given in Sect. 3.5.1, when the variation of biogenic source 1 was discussed. Furthermore, a second paper is under preparation and is dedicated to drivers of BVOCs emissions at CAO.

Correction applied in the revised manuscript (Page 15, lines 27 - 31): *"Surprisingly, α-pinene concentrations were elevated during nighttime than during the daytime. A similar nocturnal pattern has been observed elsewhere (Harrison et al., 2001; Kalabokas et al., 1997; Kalogridis et al., 2014) and was attributed to nocturnal emissions from monoterpenes storing plants from the understorey vegetation. These nocturnal maxima were also enhanced by the slow removal processes (i.e. low concentrations of oxidizing species) and the shallow nocturnal boundary layer."*

**6/ Page 20, "3.5.3 Regional background (factor 6)":** It would be suggested to add a clear definition of "regional background". The definition would be helpful to understand the factor 6, since the source areas associated with factor 4, 5 and 6 are all South of Turkey or Southwest/Southeast of Turkey.

As noticed by reviewer #2, CF plots highlighted similar potential source areas associated with factor 4, 5 and 6. However, contributions of these factors were different in function of air masses origin as depicted in Fig. 16 of the revised manuscript.

Contributions of factor 6 were rather stable whatever the air mass origin (from 5.2 µg.m$^{-3}$ to 6.6 µg.m$^{-3}$) that underlines the background character of this factor. This factor was mainly composed of ethane, propane and some OVOCs, species with high atmospheric residence times. Because of their low reactivity, species of this factor tend to accumulate in the atmosphere and show significant background levels. Factor 6 can be interpreted as a regional contribution of various remote sources of the Mediterranean region transported towards the receptor site by aged air masses which have not been recently in contact with relatively significant anthropogenic sources.

Contrarily, factor 4 and 5 were more influenced by regional contributions. Factors 4 and 5 contributions were twice as high when the station was under continental influence comparing to the one under marine influence (Fig. 16 of the revised manuscript). Furthermore, the authors noticed a higher continental influence on VOC anthropogenic factors contribution when air masses were originated from the Eastern Mediterranean (clusters 4, 5 and 7 associated to air masses originating from West Asia) compared to the Western Mediterranean (cluster 3 associated to air masses originating from Europe) as expected with the distance of respective potential emission areas. The strongest continental influence was when air masses originated from the Eastern Mediterranean region. As a result, factors 4 and 5 were of local/regional origins but were not representative of the continental regional background as factor 6.

As suggested by reviewer #2, a definition of "regional background" was added in the revised manuscript (Pages 20 - 21, lines 29 - 2): *"As a conclusion, factor 6 can be interpreted as a regional contribution of various remote sources of the Mediterranean region, showing hence the continental regional background (Hellén et al., 2003; Leuchner et al., 2015; Sauvage et al., 2009). These sources were transported towards the receptor site by aged air masses which have not been recently in contact with additional anthropogenic sources. Within the time of transport of emissions from distant sources, atmospheric oxidation removes the reactive species and the remaining fraction contains mostly the less-reactive species, such as ethane, propane and some OVOCs. Finally, it is reported here as "regional background"."*

**7/ Pages 23 – 25, "4.2.2 Relationship between VOCs and OA":** It would be suggested to add p value associated with correlation coefficient (r). With the p value, it would be more convincing to state that the correlation is statistically significant.

As suggested by reviewer #2, p-values associated to each Pearson correlation coefficients of the Section 4.2.2 (Pages 24 - 25) were calculated and indicated in the

revised version of the manuscript. The authors stated that Pearson correlations discussed in this section were statically significant since all p-value were below 1.3 10$^{-14}$.

**8/ Page 46, figure 5:** the word "anthropogenic" in the caption of Figure 5 would be "biogenic"; the word "m69" in the fourth drawing would be "isoprene".
The word "anthropogenic" in the caption of figure 5 was replaced by "biogenic" and the word "m69" in the fourth drawing was replaced by "isoprene".